# Fast widefield scan provides tunable and uniform illumination optimizing super-resolution microscopy on large fields

Adrien Mau [1,2], Karoline Friedl [2,3], Christophe Leterrier [3], Nicolas Bourg [2] & Sandrine Lévêque-Fort [1✉]

Non-uniform illumination limits quantitative analyses of fluorescence imaging techniques. In particular, single molecule localization microscopy (SMLM) relies on high irradiances, but conventional Gaussian-shaped laser illumination restricts the usable field of view to around 40 µm × 40 µm. We present Adaptable Scanning for Tunable Excitation Regions (ASTER), a versatile illumination technique that generates uniform and adaptable illumination. ASTER is also highly compatible with optical sectioning techniques such as total internal reflection fluorescence (TIRF). For SMLM, ASTER delivers homogeneous blinking kinetics at reasonable laser power over fields-of-view up to 200 µm × 200 µm. We demonstrate that ASTER improves clustering analysis and nanoscopic size measurements by imaging nanorulers, microtubules and clathrin-coated pits in COS-7 cells, and β2-spectrin in neurons. ASTER's sharp and quantitative illumination paves the way for high-throughput quantification of biological structures and processes in classical and super-resolution fluorescence microscopies.

[1] Institut des Sciences Moléculaires d'Orsay, Université Paris-Saclay, CNRS, Orsay, France. [2] Abbelight, Cachan, France. [3] Aix-Marseille Université, CNRS, INP UMR7051, NeuroCyto, Marseille, France. ✉email: sandrine.leveque-fort@universite-paris-saclay.fr

In advanced widefield fluorescence microscopy, lasers are a common excitation source: they provide excitation at precise wavelengths coupled with high control over the delivered power, both properties that are crucial to obtain quantifiable images. Typically, the laser is focused at the back focal plane (BFP) of the objective to produce a large, collimated beam illuminating the whole sample. While widefield fluorescence microscopy is a fast imaging method, resulting images are usually contaminated by blur from below and above the plane of focus, clouding the fluorescence signal.

Optical sectioning improves the signal by spatially limiting the illumination around the focal plane: highly inclined and laminated optical sheet (HiLo) excitation translates the laser beam in the BFP to an oblique illumination of the sample[1]. Placing the beam at the position corresponding to illumination at the critical angle results in a "grazing incidence", ~1-μm thick illumination sheet above the coverslip[2,3]. Inclining the beam further[4] results in total internal reflection fluorescence (TIRF)[5], restraining the illumination to an exponentially decreasing intensity over a few tens of nanometers above the coverslip surface. These remarkable sectioning capabilities can be performed on one single setup[6–8] and allow to study membrane and adhesion processes with minimal background. In practice, however, TIRF suffers from heterogeneous illumination caused by the interference patterns arising from the high spatial coherence of lasers and scattering. Rapidly spinning the beam around the BFP can alleviate these fringes by averaging beam orientations over a single camera frame[9], a method since applied with several variants and refinements[10–12].

The methods above result in Gaussian-shaped illumination profiles over the sample. This is sufficient for the typical field of view (FOV) acquired by EMCCD cameras but is more problematic over larger FOVs acquired by newer, highly sensitive sCMOS cameras[13]. The non-uniformity of Gaussian-shaped illumination lowers exploitable FOV sizes and thereby decreases the imaging throughput, a significant caveat for quantitative analysis of images obtained by TIRF.

The need for uniform excitation is even more pressing in single-molecule localization microscopy (SMLM) such as (f) PALM[14,15] or PAINT[16] where the localization precision strongly varies with the number of emitted photons. It is even more considerable in (d)STORM[17–19], where the single-molecule regime ($\ll$1 emitting molecule/$\mu m^3$) relies on driving most fluorophores in a dark state, provided through a high irradiance (kW/cm$^2$). As the transition to the dark state is highly dependent on the local excitation intensity, non-uniformities of illumination result in a strongly heterogeneous blinking behavior and loss of image quality across the FOV. For all SMLM methods, no matter the origin of the single-molecule emission, non-uniform precision precludes proper analysis of SMLM images over large fields of view.

Thus, several recent studies have aimed at obtaining a uniform excitation over a large FOV. For example, waveguides[20–22] provide excellent fixed TIRF on large fields, but cannot be restricted to the actual FOV acquired by the camera, illuminating and bleaching the whole sample at once. Classical solutions revolve around beam-reshapers[23–26] and multimode fibers[27–29] but are also restrained in field adaptability. Spatial light modulators (piSMLM[30]) may adapt the shape and size of the FOV but suffer from high-power loss and are rather expensive and complex. All these classical methods illuminate the whole field at once, so they may be ill-adapted to TIRF (see Supplementary Note 1), need speckle reducers and provide larger FOVs under the premise of using higher laser power. Additionally, focusing a high-power laser beam at the edge of the BFP may damage the lens at the back of the objective.

To circumvent the compromise between laser power requirements, optical sectioning performances, and field uniformity, we developed Adaptable Scanning for Tunable Excitation Region (ASTER). ASTER is a hybrid scanning and widefield excitation scheme that can perform epifluorescence, oblique, or TIRF illumination, while providing illumination uniformity at variable FOV sizes adapted to the camera or sample. Being a general widefield illumination scheme, ASTER can benefit to both classical widefield fluorescence microscopy and SMLM.

## Results

**Flat-top epifluorescence/TIRF excitation principle.** ASTER is a hybrid scanning and widefield excitation scheme. Any classical widefield setup can be converted into ASTER configuration by smoothly integrating alternative optical conjugation along with a scanning device such as galvanometers. In our implementation (Fig. 1a), the initial Gaussian beam, which provides a limited and non-uniform excitation, is focalized between two galvanometer scanning mirrors placed in a plane conjugated to the BFP of the objective so that an angle shift applied to the mirrors will induce a similar angle shift in the objective BFP and a position shift of the beam at sample plane. This configuration allows for large $X$–$Y$ area scans of a collimated beam. Fast scanning of the Gaussian beam position in defined patterns such as raster scan or an Archimedes spiral then generates an overall homogeneous illumination (Supplementary Fig. 1) over the FOV when averaged over the camera frame exposure time. The field size can be increased or diminished in milliseconds without physical intervention by adapting the galvanometer input amplitude. Notably, as the polar angle of the beam varies at the BFP while its position is maintained, this flat-top excitation scheme is compatible with inclined illumination such as oblique or TIRF. To this end, a conventional motorized translation stage serves as switch from epifluorescence to oblique and TIRF excitation.

To generate a uniform flat-top excitation over the whole FOV, the scanning needs to meet two criteria. First, the maximum distance between adjacent lines on the beam path has to be lower than $1.7\sigma$ (Supplementary Fig. 1), $\sigma$ being the standard deviation of the input Gaussian excitation beam. Interestingly, decreasing that gap will not affect the flat-top so that smaller gaps may be used. For a given field size, this spatial rule defines the minimum number of lines needed to achieve homogeneity (Supplementary Note 2). Second, to avoid stroboscopic effects the flat-top must be synthetized under a scanning period Tscan that divides the camera integration time Tint (Fig. 1d). A typical galvanometer mirror has a repositioning delay of 300 μs, so the number of scanned lines will set the minimal required time to synthesize the flat-top profile. Our implementation uses an input excitation beam of $\sigma = 17$ μm and gaps between 1.2 and $1.4\sigma$: Ten lines are sufficient to generate a flat-top profile on a 200 μm × 200 μm FOV under 5 ms, which is two times the maximum frame rate of classical sCMOS (100 fps) cameras. In practice, we used camera integration times between 50 and 100 ms and a scanning period of half the integration time so that the flat-top was averaged twice over a single frame, though this number may be modified by adapting the scanning period, so that the flat-top is averaged either once, or multiple times (Supplementary Fig. 2). Notably, compared to confocal laser scanning ASTER does not perform point scanning but a continuous scan with a wide input collimated beam and thus can cover large areas much faster.

To characterize the illumination homogeneity and validate our simulations, we imaged a thin layer of fluorescent Nile Blue (Fig. 1b, c) with a classical wide Gaussian beam excitation ($\sigma = 45$ μm) and with our ASTER illumination scanning a raster pattern of 150 μm long lines ($\sigma = 17$ μm). Figure 1c shows that the

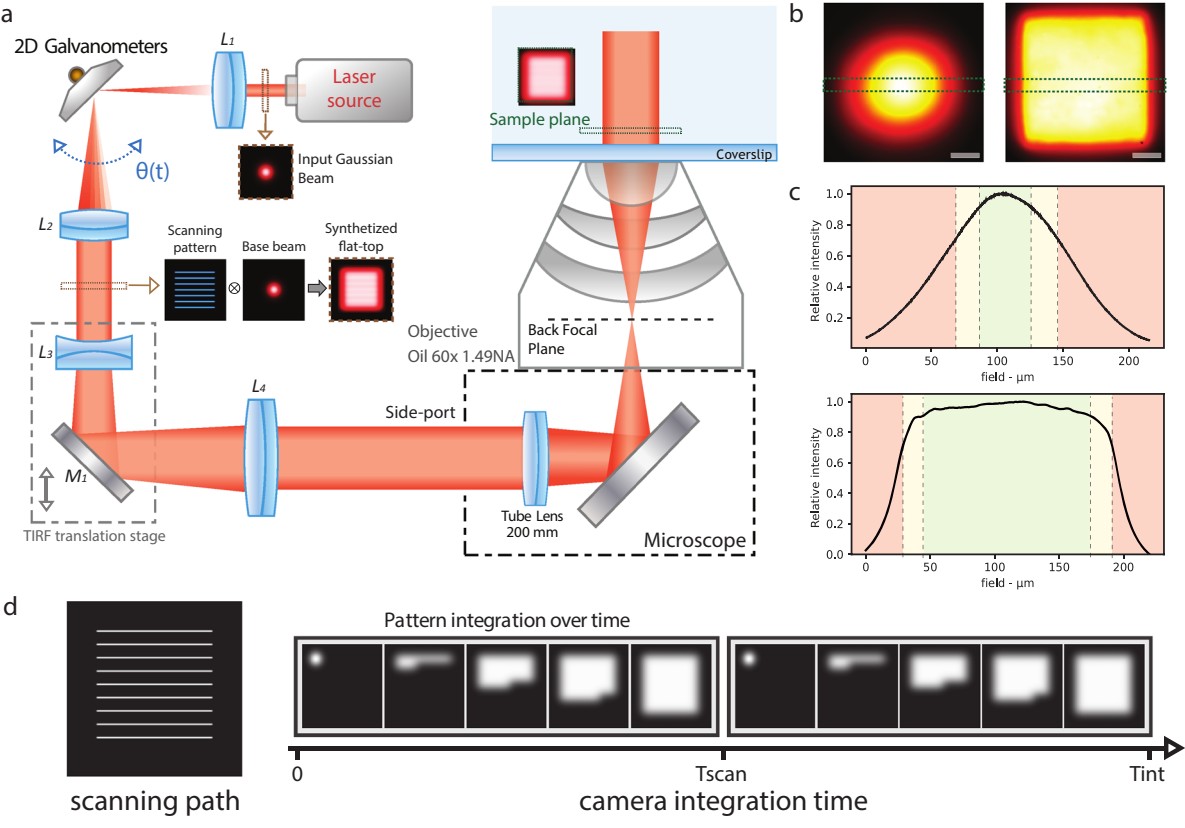

**Fig. 1 Schematic of ASTER and resulting illumination patterns. a** Simplified schematic of ASTER setup generating a homogeneous field using a raster scanning pattern. $L_i$ are lenses with focal length fi: f1 = 100, f2 = 100, f3 = −35, f4 = 250. $M_1$ is a dielectric mirror. A small input Gaussian base beam is scanned in-between the $L_1$ and $L_2$ lenses, resulting in a collimated flat-top profile, which then goes through a TIRF translation stage and is magnified between $L_3$ and $L_4$. After focalization at the BFP of an objective lens, it results in a temporally averaged flat-top excitation profile at the sample. **b** Thin layer of fluorescent Nile Blue imaged at low laser power with a fixed Gaussian excitation beam (left) and with ASTER (right) raster scanning excitation. Scalebars 40 μm. **c** intensity profiles from **b** of Gaussian (up) and ASTER (bottom) illuminations taken along the green dashed area. Dashed lines and colors indicate the field ranges in which intensity is above 90% (green), between 70 and 90% (yellow) or below 70% (red) of its maximum value. **d** Scanning path (left) and generation of the uniform profile over temporal acquisition of the camera. (right). With Tint the camera integration time, the example of a scanning period Tscan = Tint/2 is shown.

ASTER illumination triggers homogeneous Nile Blue fluorescence over a single camera frame, with a square shape matched to typical camera detectors. The resulting flat-top illumination profile is consistent with our simulations and exhibits significant flatness over ~130 μm, which could be diminished or increased by adapting the galvanometer input amplitude. In this configuration, if we consider that intensity should remain over 90% of its maximal value for confident quantification over the FOV, the Gaussian illumination would be limited to a 32 μm × 32 μm usable FOV, while ASTER can provide at least a ~16X larger, 130 μm × 130 μm FOV. On Fig. 1c ASTER exhibits Gaussian-shaped borders that reflects the use of a base Gaussian beam of $\sigma = 17$ μm. A smaller base Gaussian beam may be used to sharpen the flat-top borders, but at the cost of slower imaging speed as more lines will have to be scanned. The decrease in brightness at the periphery of the image also stems from vignetting, an effect occurring on all microscope objectives[31] as light beams emanating from the periphery of the field are partially blocked by optical or mechanical components. We confirm this phenomenon by scanning a large flat-top illuminating the full field of the camera (Supplementary Fig. 3). With our 60X magnification and square fields, vignetting is negligible for fields smaller than 160 μm × 160 μm, at 200 μm × 200 μm up to 21% intensity is lost at the corners (affecting 9% of the field), this increases up to 35% loss of intensity on the full 220 μm × 220 μm field of our camera (affecting 18% of

the field). In conclusion, even though a wide uniform excitation can be provided, homogeneity is ultimately limited by detection to uniform fields of 160 μm × 160 μm, and relatively uniform fields of 200 μm × 200 μm. By working on a circular field, however, a vignetting-free area of 200 μm × 200 μm (radius of 113 μm) can be specifically illuminated by scanning an Archimedes spiral (Supplementary Fig. 1).

We then assessed the compatibility of ASTER with inclined, optically sectioning illumination schemes, where a precise alignment and focusing of the excitation beam in the BFP of the objective is crucial (Supplementary Fig. 4). We focused on TIRF, as it is one of the most common schemes used in SMLM. First, we compared TIRF to classical epifluorescence illumination (EPI) obtained through ASTER by imaging 3-μm diameter beads, coated with biotin, and labeled with AF647-streptavidin. As can be assessed on Supplementary Fig. 5, due to the spherical shape of the beads the experimental sectioning depth is reflected through the beads apparent size[32,33]: imaging of beads from EPI to TIRF resulted in the shrinking of the beads radii. This effective sectioning stems from both optical sectioning of the illumination and depth of field of our objective. We then imaged beads on a 160 μm × 160 μm FOV and measured their respective size in EPI and TIRF (Fig. 2a, b). To assess the existence of spatial correlation, we measured the sectioning depth of each individual sphere that we define as twice the height of the intensity ring. The

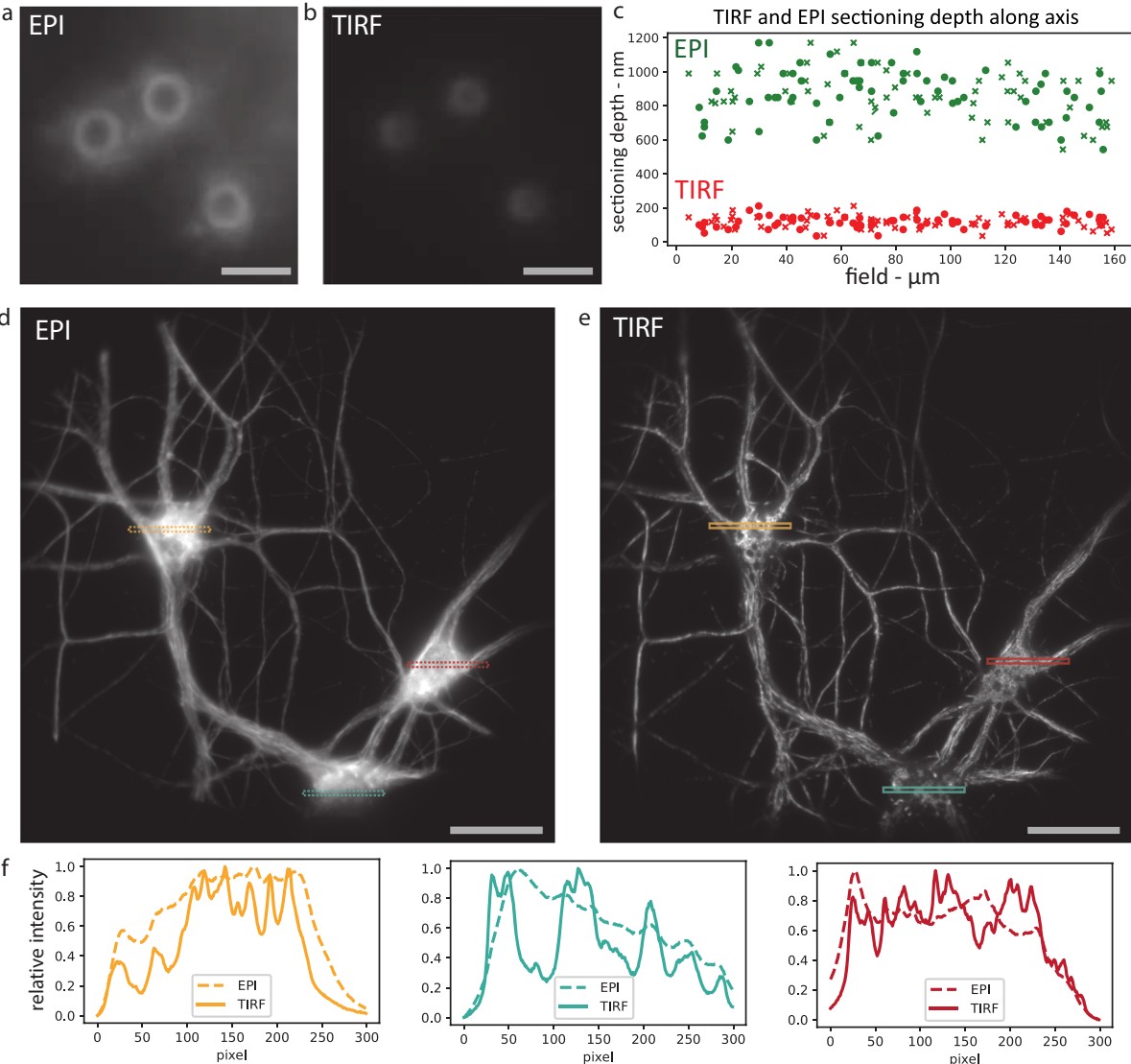

**Fig. 2 ASTER TIRF illumination.** Illumination of 3 μm beads with focus at the coverslip, in ASTER epifluorescence (EPI, **a**) and ASTER TIRF (**b**) illuminations (whole images are shown in Supplementary Fig. 6). Scalebars 4 μm. **c** Measured sectioning depth for 67 individual beads in EPI (green) and TIRF (red) illuminations on a large 160 μm × 160 μm FOV. Cross and circle markers, respectively, denote measurement along the *x* and *y* axis of the sample plane. In total, 200 μm × 200 μm imaging FOV of neurons labeled with an anti-β2-spectrin primary and an AF647-coupled secondary antibody, illuminated with raster scanning ASTER, with a scanning period of 50 ms and an exposure time of 100 ms, in either classical epifluorescence (**d**) or TIRF (**e**) illumination schemes. Scalebars 40 μm. **f** Normalized EPI and TIRF profiles of each colored area in **d** and **e**.

measured sectioning depth for TIRF excitation is $117 \pm 35$ nm (mean ± standard deviation), while epifluorescence yields a depth of $865 \pm 149$ nm, likely defined by the objective's depth of field (Fig. 2c). The sectioning depths for both schemes are uniform over the FOV, and their variations show no local or global spatial correlation (Supplementary Fig. 6 and Supplementary Note 3), demonstrating the absence of a spatial excitation anisotropy. Variation between beads most likely stem from both measurement precision and physical discrepancy of the bead population.

To assess the optical sectioning efficiency in biological samples, we imaged rat hippocampal neurons labeled for ß2-spectrin, a submembrane scaffold protein lining the neuronal plasma membrane, revealed with AF647. We compared EPI and TIRF configurations on a 200 μm × 200 μm FOV (Fig. 2d, e). The images show that ASTER with TIRF maintains the quality of optical sectioning along the whole FOV: fluorescence over the cell bodies of neurons (parts that are thicker than the sectioning depth) exhibit less blurred fluorescence and a better signal can be

observed compared to the epi-illuminated image, revealing the delicate structure of the neuronal network (Fig. 2f). While a 200 ms integration time was used to improve signal to noise ratio, ASTER can provide uniform TIRF excitation under 5 ms integration times (Supplementary Fig. 7), which makes it adapted to imaging fast live dynamical processes. A disadvantage of TIRF with classic Gaussian-shaped laser beams are interference patterns: TIRF with ASTER, by contrast, exhibits no such inhomogeneous patterns (Supplementary Fig. 8), as they are likely to be averaged out by beam scanning and camera integration. Even though ASTER is still subject to shadowing effects, it solves both the issues of TIRF interference fringes and nonuniform Gaussian illumination, with the benefit of a large achievable FOV. In conclusion, ASTER leads to TIRF images with a similar quality as spinning azimuthal TIRF, with the added benefit of field uniformity and FOV size versatility. ASTER illumination efficiently provides both uniform spatial illumination and uniform axial optical sectioning for fluorescence microscopy in both EPI and TIRF illumination.

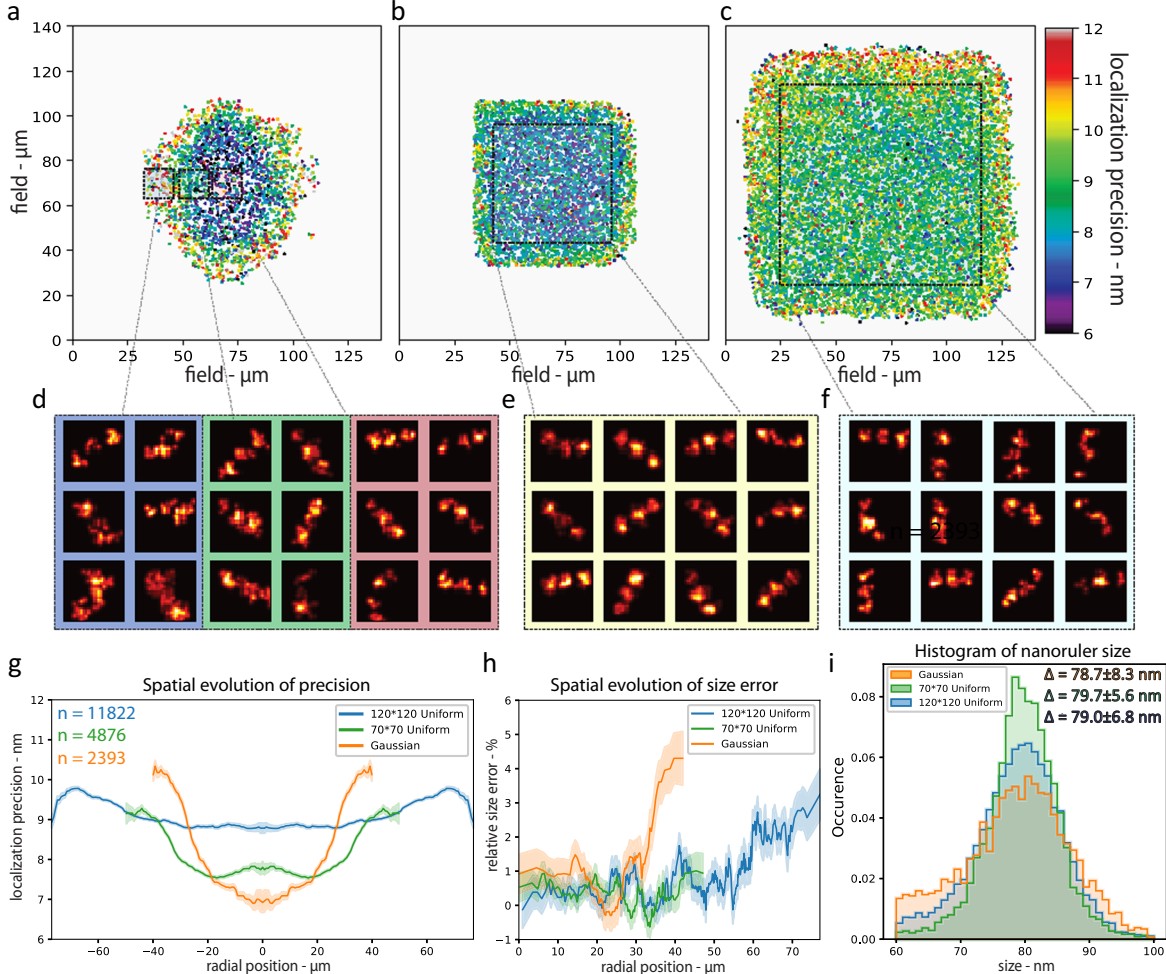

**Fig. 3 Nanorulers imaging for localization precision estimation.** DNA-PAINT imaging of 40 nm spaced 3 spots nanorulers, obtained with Gaussian (**a**, **d**), ASTER small field of view (70 μm × 70 μm, **b**, **e**), and ASTER large field of view (120 μm × 120 μm, **c**, **f**) illuminations. **a–c** are resulting localization precision maps where each point represents the average precision for one individual nanoruler (3 spots). **d–f** are nanoruler super-resolution images (150 nm × 150 nm), taken randomly from highlighted areas in **a–c**. **g** Mean localization precision along FOV radius for each excitation scheme (symmetrized). The number of nanoruler for each excitation scheme is indicated in a similar color. For each colored curve, the surrounding transparent curve indicates the standard deviation around the mean precision at a given radius. **h** Resulting size estimation error along FOV radius for each excitation scheme. A size error above 0 indicates that the nanoruler spots were measured <80 nm apart. Each colored curve is surrounded by a transparent curve that indicates the standard deviation around the mean size. **i** Resulting size measurement histogram for each excitation scheme. The mean and the standard deviation for the size (symbolized by Δ) are indicated in the upper right corner.

**Large field uniform SMLM imaging**. Next, we applied ASTER to SMLM experiments, namely DNA-Point Accumulation in Nanoscale Topography (DNA-PAINT) and STochastic Optical Reconstruction Microscopy (STORM). To assess the effect of ASTER illumination FOV size and homogeneity in SMLM experiments, and compare it to a classical Gaussian illumination, we first imaged three spots, 40 nm spaced nanorulers using DNA-PAINT (Fig. 3). Three different TIRF excitation schemes (Gaussian, $\sigma = 45$ μm), ASTER on a 70 μm × 70 μm FOV, and ASTER on 120 μm × 120 μm FOV were used, with the other parameters remaining identical. In the single-molecule regime, each of the three nanoruler spot acts as a source of blinking fluorescence, resulting in a set of localizations spread by the pointing accuracy of each blinking event. For analysis we applied the following algorithm: first, individual nanorulers were isolated by DBscan clustering[34], then for each individual nanoruler the point cloud corresponding to the three spots was fitted by a Gaussian mixture model (GMM) assuming three normal distributions. The GMM assessed the most probable mean position and standard deviation of each spot (see "Methods", Supplementary Fig. 9). For each

individual nanoruler, the mean standard deviation of all spots was then considered as a local measure of the experimental localization precision.

The Gaussian excitation resulted in a bell-shaped localization precision map (Fig. 3a, g): at the center of the FOV, the localization precision is 7 nm, but it quickly increases with the distance from the center. At 20 μm, it is 8 nm, and up to 11 nm at the edges of the FOV, 1.6 times worse than at the center (Fig. 3d, g). The mean localization precision is then 8.7 ± 1.8 nm (mean ± standard deviation). Meanwhile, ASTER excitation on a similar FOV provided a localization precision of 7.9 nm ± 0.9 nm—ranging from 7.5 to 8 nm at 30 μm from the center of the field (Fig. 3b, g). On a large 120 μm × 120 μm FOV with similar parameters, ASTER provided a 9.2 nm ± 1.1 nm localization precision (Fig. 3c, g), from 8.8 at the center of the field up to 9.5 nm at a 60 μm radial distance. This means that a 20X increase in the FOV size came at the cost of a 1.2 worse localization precision. It is conceivable that a localization precision below 9 nm could be reached by carefully optimizing imaging parameters such as laser power, optical sectioning, and camera integration time. Moreover, the inhomogeneity of the

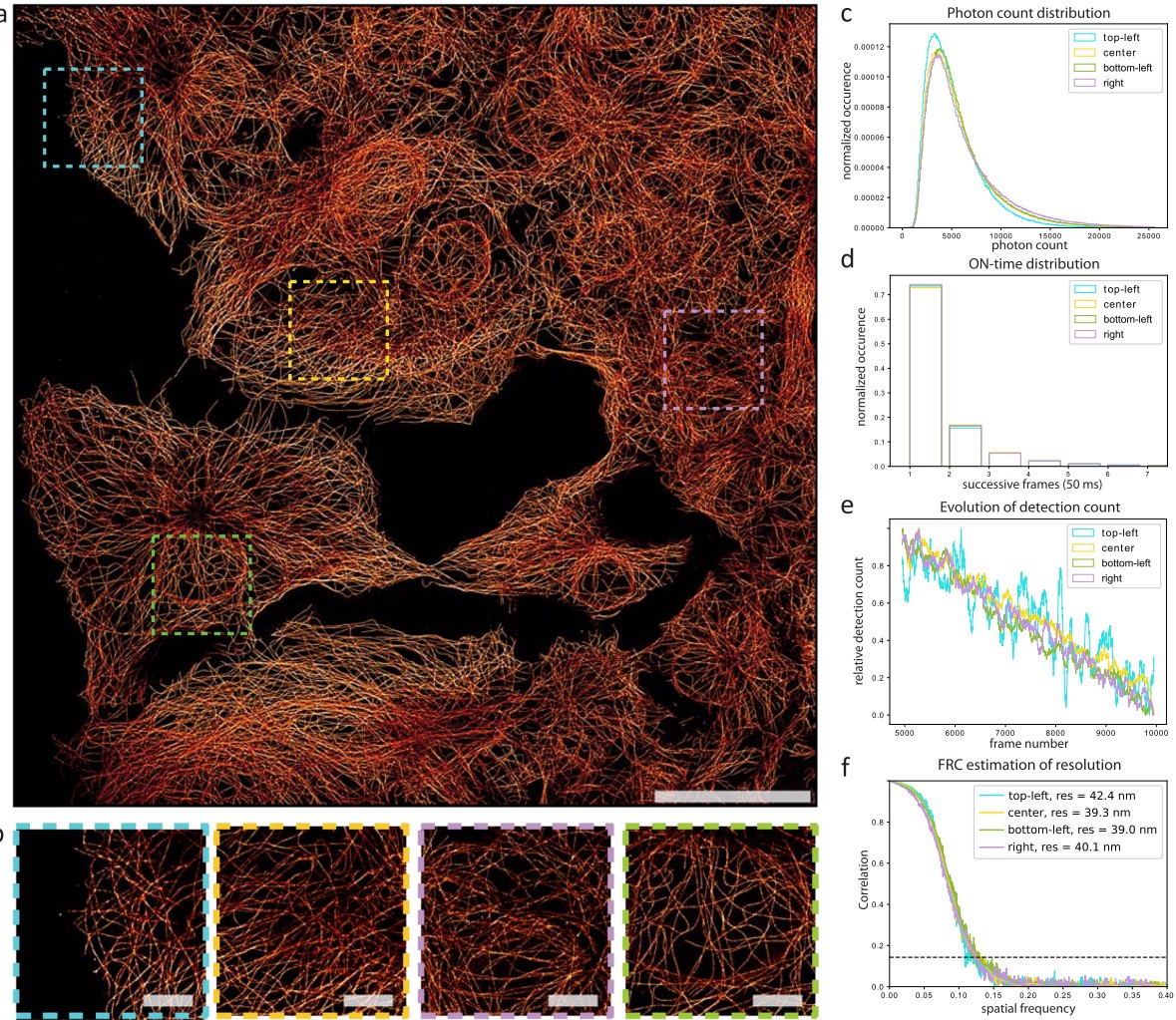

**Fig. 4 STORM imaging using ASTER. a** ASTER STORM imaging of COS-7 cells labeled for microtubules and an AF647-coupled secondary antibody, FOV size 200 μm × 200 μm, 20,000 frames at 20 fps. Excitation consisted in a ten-line scan with a laser power of 250 mW at the BFP, a gap of 1.4$\sigma$ and a 25 ms scanning period. Scalebar 50 μm. **b** Zoomed views of highlighted areas in **a**. Scalebars 10 μm. **c** Photon count distribution histogram for highlighted areas in **a**. **d** Blinking ON-time distribution for highlighted areas in **a**, expressed in number of successive frames (50 ms camera integration time). **e** Temporal evolution of detection count for highlighted areas in **a**. **f** FRC estimation of resolution for highlighted areas in **a**.

Gaussian illumination impacted the size estimation of the nanorulers along the FOV (Fig. 3h). We measured the end-to-end size of the identified nanorulers (ground truth value 80 nm). Gaussian beam illumination images yield a fairly constant relative size error of 1% at the center of FOV, rising to 3% at 35 μm from the center. ASTER provided homogeneous measurements on a wider FOV: the relative size error remained at 0.5% up to 45 μm from the center of the FOV. All the illumination conditions resulted in similar mean values for the nanoruler size (Fig. 3i), but we observed an increased number of cases where the size was underestimated to 60–70 nm with the Gaussian beam illumination, indicating a poor single-molecule regime.

We then turned to STORM experiments on biological samples. Traditionally, STORM demands strong laser power (>2 kW/cm²) to drive organic fluorophores into a blinking regime[35,36]. To induce a satisfactory blinking regime on a 200 μm × 200 μm FOV thus requires the use of 1–5 W power lasers[27].

However, as ASTER provides locally high excitation irradiance on a short time scale, and a lower global average excitation on longer scales, it may partly overcome this irradiance threshold rule. We applied ASTER in HiLo to a STORM experiment and found that even with reasonable laser power (<0.3 W at BFP),

ASTER was able to induce and maintain a densely labeled sample in the sparse single-molecule regime (<1 molecule per μm³) on large FOVs (Fig. 4), where conventional illumination would fail. It appears that the high but intermittent local excitation intensity (~12 kW/cm²) nonetheless sends most of the molecules in a long-lived dark state efficiently, as is expected for high irradiances[37].

Figure 4a shows a SMLM image of a COS-7 cell labeled for microtubules and an AF647-coupled secondary Fab₂ antibody. It was obtained using 20,000 frames and a 50 ms camera integration time, using ASTER with 0.3 W laser power, raster scanning a 200 μm × 200 μm FOV (ten lines). The microtubules are well resolved throughout the whole FOV; the zoomed images in Fig. 4b show the image quality in several different parts of the image. Analysis of these regions revealed comparable photon distributions, blinking ON-time and localization density during acquisition (Fig. 4c–e). Even though a slight decrease in photon count can be noticed at the edge, other blinking characteristics remain unchanged and suggest once again inhomogeneous detection due to vignetting at the periphery of the objective. Analysis from regions of an image acquired with a classical Gaussian illumination showed significant differences between the regions, underlining the detrimental effect of inhomogeneous Gaussian

illumination (Supplementary Fig. 10a–f). We assessed the experimental image resolution with Fourier ring correlation analysis[38] (Fig. 4f), and found that the region subject to vignetting had a close resolution (42 nm) to the other areas (39–40 nm) indicating that vignetting does not significantly impact the uniformity on our FOV. This confirms ASTER's ability to obtain uniform blinking and resolution on large 200 μm × 200 μm FOVs in STORM. Noteworthy, we do not notice artifact emerging from the temporal scanning of the beam, indicating that the position of a fluorophore relative to the scanning part does not matter. Further experiments (Supplementary Fig. 11) suggest that STORM blinking properties remain similar as long as the same mean irradiance is provided.

ASTER thus is compatible with both DNA-PAINT and STORM experiments even with typical lasers currently used on SMLM microscopes with output power below 1 W. In SMLM, because of a required pixel imaging size around 100 nm, camera chip finite size will ultimately limit the FOV, the largest uniform FOV reported so far being 221 μm × 221 μm by Zhao et al.[27]. However, their implementation did not perform TIRF and required multiple lasers with >1 W output power plus a vibration motor to reduce speckles. Generally, imperfections from the detection path will limit the maximum achievable FOV. To overcome this limit, we stitched four 160 μm × 160 μm uniform STORM images, resulting in a 300 μm × 300 μm image (Supplementary Fig. 12a, b) with minimal overlap and high uniformity. Stitching results in minimal artifacts in the overlapping edge areas, but slightly suffers from temporal effects on photon count and molecule density, mostly due to buffer consumption between acquisitions (Supplementary Fig. 12c). To limit temporal effects, one may choose to speed up STORM experiments by increasing the global irradiance[37,39]. With ASTER, this can be done by reducing the amplitude of scanning. We scanned five 25-μm long lines in 5 ms to reach an effective irradiance of 27 kW/cm², this allowed to perform fast STORM imaging of microtubule in under 100 s on a classical FOV (Supplementary Fig. 13). Such experiments are less prone to drift and highlight the practical versatility of ASTER for optimizing STORM experimental needs[40].

As ASTER homogenously illuminates large FOVs, it extends the possibility of quantitative analysis of nanoscopic structures to whole cells or group of cells. To obtain a precise view of a biological structure at the nanoscale, it is crucial to leverage the imaging of a large number of similar structures. This allows to not only obtain their average characteristics, but also the individual variation of these characteristics caused by biological variability. We imaged clathrin clusters and clathrin-coated pits by STORM in COS-7 cells (Fig. 5a–c) and applied a cluster analysis. Three COS-7 cells were imaged at once over a large 140 μm × 140 μm FOV, containing ~20,000 individual clathrin clusters. In comparison, a classical 30 μm × 30 μm FOV would have yielded ~1500 pits. The high number of clathrin clusters identified on resolution-uniform images allowed for population estimation from the characteristics of clusters. We picked specific parameters such as diameter and hollowness (see "Methods"). We were able to distinguish four populations from the cluster diameter distribution, as fitted with normal distributions (Fig. 5b). Small-diameter clusters (below 80 nm, blue and green population of Fig. 5b) likely correspond to pits in formation, while large ones (orange and red populations in Fig. 5b) are likely to be fully assembled pits. We specifically extracted large, hollow clathrin assemblies based on the ratio between the diameter and the spatial dispersion of fluorophores. Hollow clathrin assemblies with diameters of 80–200 nm would be of typical size for the large clathrin-coated pits found in fibroblasts[41]. Interestingly, some large, hollow pits showed more than one fluorescence "holes" within them, suggesting that they are either assemblies of smaller

pits or that the fenestration of clathrin cages[42] (pentagon or hexagons of 18 nm side length) can sometimes be resolved (Fig. 5c).

The large FOV provided by ASTER illumination coupled with large-chip sCMOS cameras also have interesting application for imaging neuronal cells, which grow axons over hundreds of microns in culture. Traditionally, SMLM imaging of axons has been limited to <50 μm segments of axons, impeding the visualization of rare structures and the definition of their large-scale organization[43,44]. We labeled rat hippocampal neurons for β2-spectrin, a protein that forms a periodic submembrane scaffold along axons by linking actin rings[45–47]. A 200 μm × 200 μm FOV allowed visualizing the dendrites and cell body of two neurons, and a large number of long axonal segments (Fig. 5d and Supplementary Fig. 14). The zoomed views confirm the quality and resolution of the resulting image: the periodic 190 nm organization of axonal spectrin is clearly visible, as confirmed by the corresponding Fourier transform of the images. The Fourier transform of the whole image exhibits a sharp ring at the corresponding frequency, because the banded pattern of β2-spectrins appears in axons running in all directions. On the zoomed images (Fig. 5e), the β2-spectrin along axons in one direction results in a direction-dependent frequency band on the Fourier transform, corresponding to the 190 nm spacing.

## Discussion

We implemented and characterized ASTER, a hybrid scanning and widefield illumination technique for optimized widefield fluorescence microscopy and SMLM over large FOV. ASTER generates uniform excitation over a tunable FOV without limiting acquisition speed. It has advantages over state-of-the-art uniform illumination schemes by its efficiency, flexibility, and ability to perform uniform optical sectioning schemes, such as HiLo and TIRF illuminations. We demonstrate TIRF imaging on rat hippocampal neurons on 200 μm × 200 μm, the maximal uniform FOV achievable with our ×60 magnification objective. With DNA-PAINT, we demonstrate a uniform localization precision over large FOVs (9.2 ± 1.1 nm over 120 μm × 120 μm), and even better localization precision on small FOVs (7.9 ± 0.9 nm over 70 μm × 70 μm).

ASTER also proved to be an efficient excitation method for STORM imaging experiments. Against common belief that STORM requires a strong continuous irradiance (~2 kW/cm²), ASTER induced uniform blinking dynamics at lower mean irradiance (<0.5 kW/cm²), but with a high instantaneous irradiance (~12 kW/cm²) over a large 200 μm × 200 μm FOV, alleviating the need for expensive and dangerous high-power lasers. We present biological applications by directly imaging the periodic 190 nm organization of axonal spectrin on several long axon segments from neurons. By imaging clathrin-coated pits in multiple COS-7 cells in one acquisition, we increase the number of identified clusters by a factor of 20 compared to the typical FOV of a STORM acquisition and enhance statistical analysis.

ASTER can be combined with stitching schemes, alternative objectives, adaptive detection setups, and camera chips to cover even wider FOV. Combination of ASTER with the ASOM[48,49] could be particularly interesting as hyper large fields may be imaged without moving the sample. In SMLM, the field is regularly limited to a maximum of 200 μm × 200 μm, however in classical widefield microscopy ASTER can be used with smaller magnification objectives to image larger FOVs and would be a great choice for imaging structures on larger scales. We conclude that ASTER represents a versatile and innovative tool, especially suited for SMLM. It exhibits robust uniformity and reliability, as well as adaptability to variable FOV sizes. ASTER

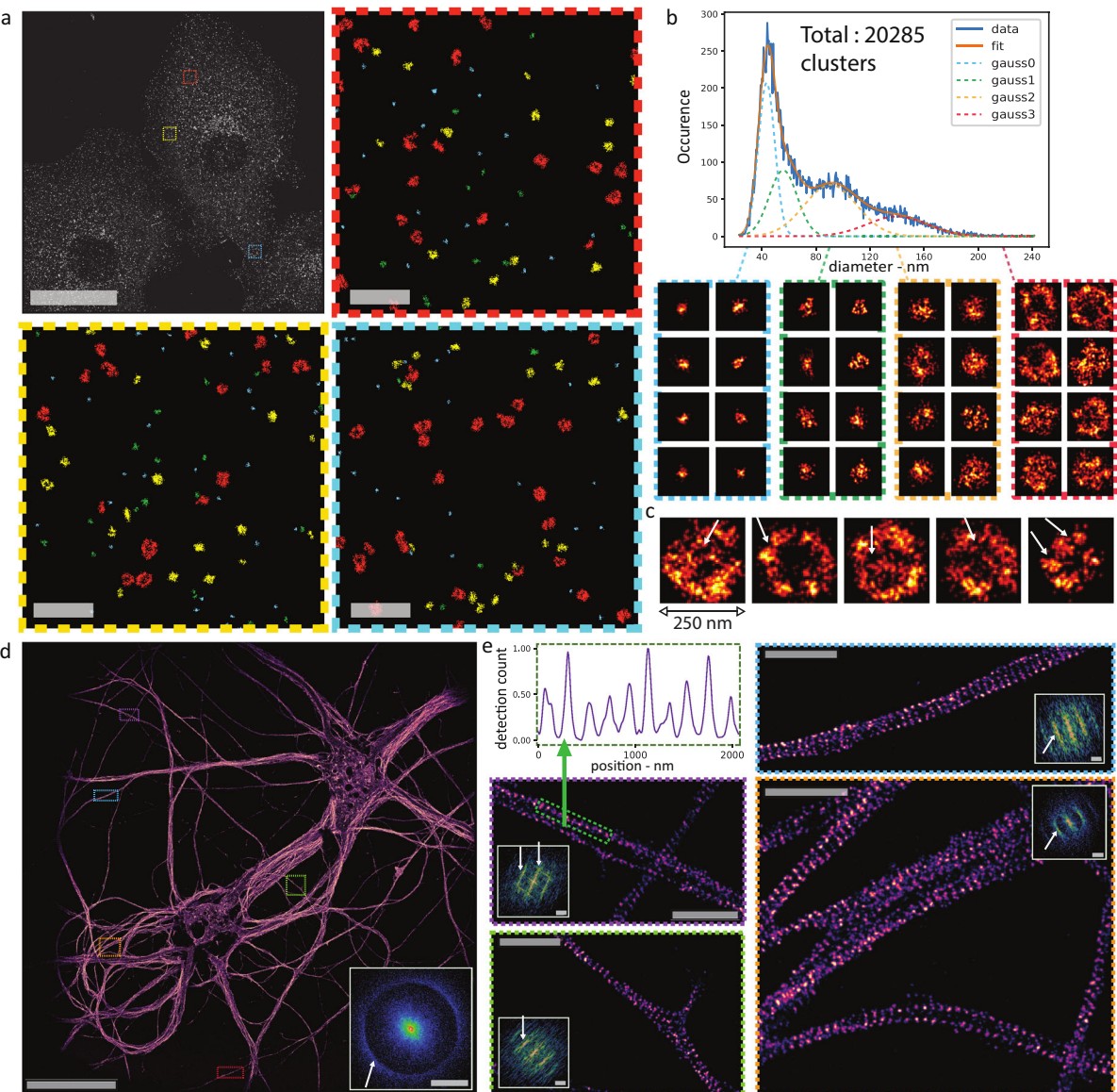

**Fig. 5 ASTER applications for single-molecule localization microscopy. a–c** ASTER STORM imaging and cluster analysis of COS-7 cells labeled for clathrin heavy-chain and an AF647-coupled secondary antibody. **a** Final 140 μm × 140 μm image with scalebar 40 μm (top left) and close-up views of the highlighted regions (colors encode cluster affiliation) with scalebars 1 μm. Pixel size is 10 nm. **b** shows the distribution of the diameter of clathrin clusters and highlights four potential populations, that can be fitted with Gaussian functions. Below are 250 nm × 250 nm images of individual clathrin related clusters, each group corresponding to a specific population. **c** shows 250 nm × 250 nm images of large, hollow clathrin clusters likely corresponding to large clathrin-coated pits. Visible cavities are highlighted by arrows. **d, e** ASTER STORM imaging and structural analysis of neurons labeled for β2-spectrin and AF647-coupled secondary antibody. **d** 200 μm × 200 μm STORM image obtained with ASTER (30 nm pixel size). Scalebar 40 μm. The two-dimensional Fourier transformation (inset) exhibits a circular frequency pattern corresponding to a ~190 nm periodicity of the staining that is present along all axons. **e** Zoomed views of regions in **d** revealing the periodic cytoskeleton along single axons (pixel size is 10 nm). Upper left image shows the intensity profile along the highlighted green line, revealing again the 190 nm periodicity. Scalebars 2 μm. For **d, e**, insets show the respective two-dimensional Fourier transformation and the known 190 nm periodicity of the axonal spectrin scaffold is highlighted by arrows. All Fourier images scalebars are 4 μm⁻¹.

may be used in combination with improved detection schemes, such as multicolor imaging or strategies to encode axial information[50,51] (see Supplementary Fig. 15). It could be beneficial to setups that modify the excitation to enhance resolution by counting photons[52–55] but ASTER implementation in such case would be challenging. The resulting uniformity may be used for demultiplexing or stoichiometry experiments[56,57], as well as in buffer characterization and other fields such as photolitography[58,59]. Finally, ASTER has potential applications in nonuniform excitation schemes, such as using smaller beams to concentrate power[60], exciting specific areas of a

sample[61] or creating a patterned irradiance on complex samples by using adaptive scanning strategies.

## Methods

**Optical setup**. We used a Nikon Eclipse Ti inverted microscope with a Nikon Perfect Focus System. The excitation was performed with an ELERA laser (638 nm) from ERROL. 6215H galvanometers from Cambridge Technology were controlled with a RIGOL DG5252 waveform generator. To maintain telecentricity, all distances between lenses are equal to the sum of their respective focal lengths. Both excitation and detection went through the left camera port of the microscope to prevent undesired cropping. To this end, the dichroic is put in front of the side port and reflects the excitation beam (not shown in Fig. 1). Fluorescence was collected

through an Olympus ×60 1.49NA oil immersion objective, a relay-system, and recorded on a 2048 × 2048 pixel sCMOS camera (Orca-Flash 4 v3, Hamamatsu). The optical pixel size was ~108 nm.

## Calibration sample preparation and imaging

*Beads*. Beads are 3 μm radius biotin-polystyrene microspheres (Kisker Biotech, PC-B-3.0) on which we attached Alexa Fluor (AF) 647 functionalized with streptavidin (Life Technologies, S21374). We prepared a solution containing 500 μl of water, 500 μl of PBS, 35 μl of microsphere solution, and 0.34 μl of AF647. This solution was centrifuged 20 min at $12,100 \times g$. The liquid was then removed and replaced with 100 μl of PBS, followed by 5 min vortexing to dissolve the deposit. In total, 50 μl of the final solution was then pipetted on to a glass coverslip and left for 20 min so that beads would have time to deposit. Finally, we added 500 μl of imaging dSTORM buffer (dSTORM smart kit, Abbelight). Images were taken at low laser power and integrated over 100 ms, for an ASTER scan period of 50 ms.

*Nanorulers*. Nanorulers (Gattaquant, PAINT-40R) consist in three aligned spots, separated by 40 nm and are labeled with ATTO655 fluorophores. To switch from ASTER to a Gaussian illumination, a constant offset was applied to galvanometer and a beam magnifier was placed between the laser and galvanometers. Parameters for imaging were chosen while optimizing the blinking with the wide Gaussian excitation ($\sigma$ = 45 μm): laser power of 200 mW, fixed TIRF configuration, and an integration time of 100 ms; parameters were maintained for each acquisition. During the acquisition on the 120 μm × 120 μm FOV, the blinking of fluorophores was slow, so fluorophores generally appeared on subsequent frames and required larger integration times or post-processing to merge them. This became apparent in the loss in resolution.

## Fluorescence immunolabeling

*Neuronal culture*. Rat hippocampal neurons in culture were prepared according to the Banker protocol[62]. Briefly, E18 Wistar rat embryo hippocampi (Janvier labs) were dissected, then cells were homogenized and plated in B27-containing Neurobasal medium on Poly-L-Lysine treated #1.5H glass coverslips (Marienfeld, VWR) to a density of 4000 cells per cm$^2$. The neurons were then co-cultured with glia cells—neuron coverslip upside down, separated from the glia on the bottom of the petri dish by wax beads. Mature neurons were fixed after 14 days in culture. All procedures followed the guidelines from European Animal Care and Use Committee (86/609/CEE) and were approved by Aix-Marseille university ethics committee (agreement D13-055-8).

Immunolabeling of neurons was performed according to recent optimized SMLM sample preparations[47,63]: neurons were fixed using 4% paraformaldehyde (Delta Microscopie, #15714) and 4% (w/v) sucrose in PEM buffer (80 mM PIPES, 2 mM MgCl$_2$, 5 mM EGTA, pH 6.8) for 20 min at RT. Cells were then rinsed with 0.1 M phosphate buffer. Blocking and permeabilization were performed in ICC buffer (0.2% (v/v) gelatin, 0.1% Triton X-100 in phosphate buffer) for 2 h on a rocking table. Primary antibodies diluted in ICC were incubated overnight at 4 °C, rinsed and incubated with the secondary antibodies diluted in ICC for 1 h at room temperature. After a final rinse with ICC and phosphate buffer, the samples were stored in phosphate buffer with 0.02 % (w/v) sodium azide before imaging. For immunolabeling, we used mouse anti β2-spectrin (BD Sciences, #612563, 2.5 μg/ml) and donkey anti-mouse AF647 (Thermo Fisher, #A31571, 6.67 μg/ml).

*Cell line culture*. COS-7 cells were grown in DMEM with 10% FBS, 1% L-glutamin, and 1% penicillin/streptomycin (Life Technologies) at 37 °C and 5% CO$_2$ in a cell culture incubator. They were plated at medium confluence on cleaned, round 25 mm diameter high resolution 1.5″ glass coverslips (Marienfeld, VWR). After 24 h, the cells were washed three times with PHEM solution (60 mM PIPES, 25 mM HEPES, 5 mM EGTA, and 2 mM Mg acetate adjusted to pH 6.9 with 1 M KOH). For preparation of STORM microtubule imaging, we added an extraction solution (0.25% Triton, 0.025% Glutaraldehyde in PEM) for 30 s then a fixation solution (0.5% glutaraldehyde, 0.5% Triton in PEM) for 12 min followed by a reduction solution (NaBH$_4$: 0.1% in PBS 1X) for 7 min. For clathrin, we directly fixed with a 4% PFA solution. Extraction and fixation solutions were pre-warmed at 37 °C. Cells were then washed three times in PBS before being blocked for 15 min in PBS + 1% BSA + 0.1% Triton. Labeling was performed in a similar solution with intermediary washing steps. α-tubulin (Sigma Aldrich, T6199) and clathrin heavy-chain (Abcam, ab2731) primary antibodies were conjugated with Rb-AF647 (Life Technologies, A21237). Cells were finally postfixed for 16 min in 3.7% formaldehyde and reduced for 10 min with NH$_4$Cl (3 mg/ml).

## Biological sample imaging

*Widefield fluorescence imaging*. TIRF imaging on neuronal sample was done at 200 ms integration times and a low 30 mW laser power. Samples consisted in β2-spectrin labeled with AF647. Output angle was adjusted with a translation stage[64] until penetration depth was roughly around 200 nm.

*STORM imaging*. STORM imaging on COS-7 cells (microtubules and clathrin) and neurons (ß2-spectrin) was performed at 50 ms exposure time using a HiLo illumination configuration. A STORM buffer (Abbelight Smart kit) was used to induce most of the molecules in a dark state. The sample was lit with laser powers of ~250 mW in the objective BFP and a scanned Gaussian beam of $\sigma$ = 17 μm. Except for clathrin, all data acquisition was excited with an ASTER excitation scanning ten lines in 25 ms and a gap of 1.4$\sigma$. For STORM on clathrin, the labeling was dense and blinking was slightly optimized by reducing the excited FOV to 140 μm × 140 μm, scanning eight lines in 25 ms with a gap of 1.2$\sigma$. The acquisitions were performed and analyzed using the Nemo software (Abbelight). Localization consisted in a wavelet segmentation after median background removal, followed by Gaussian fits of individual point spread functions. Sample was drift-corrected using a classical redundant cross-correlation algorithm.

**Image acquisition, processing, and analysis**. Neo-Live (Abbelight) was used for image acquisition. Single-molecule analysis was performed either with a homemade Python 3.7 code or with Neo-Analysis (Abbelight).

Data processing such as analysis and measurement of beads radius from Fig. 2, nanoruler from Fig. 3, and clathrin from Fig. 5 was performed in Python, whose code is available online.

*Beads*. Beads (microspheres) were detected on the TIRF image: we first applied a Laplace filter from scipy library, followed by low-pass filtering in Fourier space to diminish noise. Use of an intensity threshold then proved sufficient to efficiently detect individual beads. Peaks positions were measured via local extrema algorithms.

*Nanorulers*. Nanorulers analysis focused on resulting X, Y coordinates. A preliminary DBscan clustering was used to localize and filter out lonesome localizations. Then a more precise DBscan was used to distinguish individual groups of three-spots and associate a number to each of them. DBscan typically consider core points, which are point with at least mpts neighbor in a surrounding epsilon radius, then iteratively add adjacent points. Parameters of this secondary scan were: epsilon = 50 nm and minimum number of points mpts = 10. With these parameters, adjacent spots belonging to a similar nanoruler array were grouped together, while unwanted associations of adjacent nanorulers were minimized. For each group, a GMM was used to estimate parameters from three Gaussian distribution. GMM also estimates the mean and standard deviation of each spot, which allowed for size estimation and localization precision measurements. Nanorulers with too few points or extraordinary distance estimations were thrown away.

*Clathrin clusters*. Clathrin analysis was primarily performed via a DBscan clustering, with an epsilon parameter of 35 nm and a minimum number of points of 25. This clustering method localized each individual cluster of close points. For each of these cluster we calculated several parameters, such as the mean position and the effective diameter, Feret's diameter[65], the hollowness, the angle of orientation, and eccentricity. Effective diameter and mean position were calculated by minimizing radial dispersion among points. Hollowness consisted in the ratio between the mean radius value, divided by the standard deviation of radius. Among all parameters, the diameter and the hollowness proved to be the most relevant in term of describing cluster distributions.

*Neurons*. Fourier transform was performed on 2D histogram images via the fft2 function from the numpy.fft library of Python.

**Reproducibility**. Experiments with Nile Blue have been repeated five times, and yield similar results (Fig. 1b). When investigating the presence of vignetting, full field images of Nile Blue samples have been performed three times (Supplementary Fig. 3).

The effect of optical sectioning on beads was observed multiple times ($n > 5$) on three different samples (Supplementary Figs. 5 and 6). Optical imaging from EPI to TIRF, of neurons or other biological components, is extremely consistent: it has been performed multiple times ($n > 20$) on different setups (Fig. 2). The ability of our implementation to perform TIRF images with a scanning period of 5 ms has been repeated four times on the same sample, and should extend to any setup using similar scanning devices (Supplementary Fig. 7). Investigation of the difference between classical TIRF, ASTER TIRF, and spinning TIRF images has been realized three times (Supplementary Fig. 8).

Concerning SMLM Experiments, nanoruler PAINT experiments have been repeated three times for similar illumination configuration as presented in the manuscript, and yielded similar results (Fig. 3).

In total, 200 μm × 200 μm direct imaging in STORM has been repeated at least eight times on different samples (Fig. 4). Cluster analyses of clathrin has been performed on two samples, only one of which is shown in the manuscript (Fig. 5a–c). Wide FOV imaging of neurons has been repeated five times (Fig. 5d and Supplementary Fig. 14).

For comparison with ASTER, Gaussian illumination in STORM has been performed twice (Supplementary Fig. 10). We have performed two stitching experiments, only one of which is shown in the supplementary (Supplementary Fig. 12). Concerning the ability to perform fast STORM imaging, it has been

repeated consistently on two different setups and numerous samples ($n > 10$) (Supplementary Fig. 13).

Investigation of the effect of the scanning speed on STORM blinking kinetics has been performed once (Supplementary Fig. 11). 3D Images of microtubules on a $200\,\mu m \times 200\,\mu m^2$ FOV has been performed thrice (Supplementary Fig. 15).

**Reporting summary**. Further information on research design is available in the Nature Research Reporting Summary linked to this article.

## Data availability
Data that support the findings of this study have been deposited on Zenodo (DOI: 10.5281/zenodo.4625796), with the exception of SMLM large raw data files (>20Go), which are available from the corresponding author on reasonable request.

## Code availability
Code is available online on Github at the following link: https://github.com/AdrienMau/ASTER_code and Zenodo (https://doi.org/10.5281/zenodo.4625796).

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

## Acknowledgements

We acknowledge people who provided valuable advice and support in the course of this study, particularly P. Jouchet and C. Cabriel, as well as the CPBM/ISMO-UMR8214 for access to cell culture facility. We thank S. Vassilopoulos for discussions about clathrin organization in cells. M. Bardou and F. Boroni-Rueda helped with cell culture and labeling. C. Hubert and E. Fort provided the lasers and galvanometer scanners.

## Author contributions

A.M., N.B., and S.L.-F. conceived the project. All authors contributed to the writing and editing of the manuscript. K.F. and C.L. provided neuronal samples. Experimental acquisitions, analyses, and illustrations were performed by A.M.

## Competing interests

N.B. and S.L.F. are shareholders in Abbelight. Other authors declare no competing interests.
