## [Peer Review File · Nature Communications]

REVIEWER COMMENTS

Reviewer #1 (Remarks to the Author):

In this manuscript Adrien Mau et al introduce a new illumination method, called ASTER, for achieving SMLM with large field of view. The authors claim that ASTER is capable of providing uniform illumination without suffering the limitations from reported methods (e.g. laser damage to objective lens, power loss, or bleaching the whole sample at once). The method reported in this manuscript seems promising, but the current evaluation is insufficient, and the manuscript is loosely written. I cannot determine the performance of the method relative to others.

My major concern is that, since the ASTER method is based on 2D scanning of a Gaussian beam, the total time required to scan the full field of view (FOV) would set a hard limit for the minimum camera integration time, which has a direct impact on the temporal resolution of STORM imaging. Actually, the camera integration time used in the manuscript is typically 50-100 ms, which is OK for DNA-PAINT, but may be too long for conventional STORM experiments (where researchers would like to use 10 - 20 ms to increase the temporal resolution). The relatively long camera integration time would surely limit the spread of the ASTER method in a wider application field. The authors may consider a multi-beam strategy to break this limitation. Anyway, more works on pushing the minimal camera integration time should be done to justify a publication in a high impact journal, such as Nature Communications.

Other concerns or comments:

Fig 1: The authors mention a temporally averaged flat-top excitation profile. Any further characterization on the time-dependent behavior of the illumination? Fig 1d shows a slightly rotated square, and Fig 1e shows slopes in the illumination profile. Both of them are not desired for further image stitching. What are the origin of these and any ways to get rid of them?

Fig. 2. The authors demonstrate epifluorescence and TIRF imaging of a large FOV of 200 μm x 200 μm , but provide no details on how to enable such a large FOV. And, what is the relationship between the minimum camera integration time and the corresponding FOV? Fig 2d-e should be analyzed quantitatively, rather than presenting only those pretty images.

Fig 3: What method was used to calculate the localization precision?

Fig 4: It would be more convincing to include a highlighted area in the corner of the image.

Page 9, the last paragraph. The authors claim that they achieve the largest FOV of 200 μm x 200 μm . Actually, an even larger FOV (221 μm x 221 μm) was reported in reference 24.

Page 10, the first paragraph. The calculated irradiance of 0.5 kW/cm^2 may be misleading. Since this is basically a point scanning technique, it is better to report the intensity of the Gaussian beam on the sample plane.

Page 12. The integration time in TIRF imaging was 200 ms. Why did the authors use such a long integration time?

Supplementary Fig 7. It would be good to see statics similar to those shown in Fig 4d-f.

It is confusing to see the different FOVs in the SMLM images: 120 μm x 120 μm in Fig 3c, 200 μm x 200 μm in Fig 4, 140 μm x 140 μm in Fig 5a, 200 μm x 200 μm in Fig 5d, etc. Why did this happen?

The authors present only 2D imaging results. It would be nice to see the performance (at least discussions) of this ASTER method in 3D STORM.

The authors should revise the manuscript carefully to enable a brief and precise writing. For example, in the last paragraph in page 9, the term "SMLM PAINT" looks odd, and the last sentence should be deleted.

Reviewer #2 (Remarks to the Author):

This paper claims the novel development of scanned widefield illumination for use in epi-fluorescence, HiLo and TIRF microscopy. The scanned widefield illumination is claimed to produce a larger, more uniform FOV, using lower laser powers, than has otherwise been possible.

However, the method of scanning a wide illumination beam, either for transmitted light or fluorescence microscopy is already known as adaptive scanning [Benjamin Potsaid, Fern P. Finger, John Ting-Yung Wen, "Living organism imaging with the adaptive scanning optical microscope (ASOM)," Proc. SPIE 6441, Imaging, Manipulation, and Analysis of Biomolecules, Cells, and Tissues V, 64411D (19 February 2007); <https://doi.org/10.1117/12.699552>]. I would expect this to be referenced, and perhaps the Optics Express 2005 paper preceding it, which presented adaptive scanning but did not include fluorescence. Therefore some of the claims on novelty would need to be removed or modified.

There are also TIRF illumination scanning schemes used for structured illumination microscopy (TIRF-SIM) and single molecule localization microscopy (SMLM) (e.g. SIMFLUX, SIMPLE), and the editors may consider it relevant to present the novelty and advantages of the authors' approach in reference to these. These existing approaches involve scanning and are more complicated than the authors' approach, but in them scanning is over smaller distances using different methods and does not address the same problems.

The application of adaptive scanning for wide and uniform fluorescence excitation in TIRF and SMLM may be considered novel and will be of interest to others in the field. The paper presents promising results on precision uniformity in SMLM and large FOV and stitched SMLM images. It may lead to flexible, higher throughput data acquisition with less specialized/expensive equipment in TIRF-SMLM, which would improve or accelerate experimental results. Further, quantitative comparison with the flat-field epi-fluorescence SMLM schemes of Douglass et al. (already referenced) and piSMLM would be useful to explain any advantages in epi-fluorescence.

This work may also lead to acquisition of data from larger samples without requiring image stitching, or requiring less image stitching. Particularly, if the claims of equivalent localization data quality but with lower laser power are robust, labs that require large FOV/high-throughput SMLM data may be interested in applying this technique themselves.

An example of HiLo imaging with the larger or more uniform FOV would be useful for illustrating the claim that the method is useful here.

Discussion of the effect of modulating the excitation of single molecules would be welcome: In ASTER SMLM, are potential photons not acquired because scanning the illumination moves away from an emitting molecule before it has finished emitting?

However, more rigorous presentation of results is required in the text and supplementary material in order to justify the claims of these developments for TIRF and SMLM. For instance:

- The choice of $1.2 \cdot \sigma$ for the upper limit on the distance between scan lines is not illustrated or derived, including in Suppl. Fig. 1.
- Confidence intervals on results are needed in the main text in many places.
- Detail of the calculation of the TIRF penetration depth in supplementary would be very useful.
- Quantitative results for spatial correlation and anisotropy are required when these are mentioned

for the TIRF field.

- A limit on measurement precision is mentioned as explaining variability in results per bead when measuring the TIRF field. What is the source of this and what is its magnitude?
- More precise comparison of the images in Fig. 2d,e is needed. Fig. 2d,e may currently be mislabelled in the figure legend.
- Showing the laser speckle/interference that ASTER helps to deal with would be informative in supplementary material.
- It is claimed that the TIRF images obtained are comparable with a spinning azimuthal TIRF system, but no evidence is provided for this.
- What are the sizes of the Gaussian beams used for Gaussian and ASTER illumination in the different fields in Fig. 3?
- Quantitative discussion of the Fig. 3 results needs attention in the text. The uniformity of the ASTER results comes across as over-represented: precision is not uniformly 7.6 nm or 9 nm according to Fig. 2g. The range of mean values over the different positions, or confidence intervals would be much more useful. The non-uniformity of the Gaussian results also comes across as over-represented in discussion of Fig. 2h, where the size error is fairly uniform up to about 30 μm out from the centre. There is obviously a real difference between the scheme, but it does not seem to have been presented correctly in the text.
- It is curious that the Gaussian distribution had a smaller mean precision at the centre of the FOV, but a larger ruler size error. Why might that be?
- Suppl. Fig. 6 contains low quality images that are hard to read and therefore hard to interpret in places. Presenting a statistic like median photon counts to go with panel c might be informative. Panel g is not explained.
- What was the Gaussian beam size used in ASTER for Fig. 5, and what is the inter-line spacing? (My guess is 20 μm spacing, but I am not sure.)
- Where is the number $\sim 1,500$ clathrin-coated pits obtained from for a classical 30 μm x 30 μm FOV?
- The holes suggesting 18 nm side-length fenestrations are not currently clear in Fig. 5. Does the estimated localization precision achieved suggest that these would be resolvable in images? More information on scale in b and c is needed and some higher-magnification images may help.

Discussion:

Paragraph 1: "uniform excitation over a tunable FOV" would be sufficient and better: every field has 95% uniformity out to a certain distance, even if it smaller.

"has advantages over" in these areas would be better than "outperforms", since few quantitative performance comparisons are made.

Paragraph 2 and 3 seem unnecessary as quite a long repetition of previous results (in which the localization precision claimed in the ASTER fields is given as even lower, 7 nm, than in the results section, 7.6 nm, which also needed addressing).

Paragraph 4: Robust reliability is claimed, which is not evidenced (and sounds a little tautologous). The particular connection between ASTER and ref. 44 is not clear. I would say "ASTER may be used in combination with improved detection schemes [45, 46]", rather than that they are further uses of ASTER. The relevance of flat-fielding to photolithography should at least be referenced.

Citations: 31 and 32 are surprising here. If it is the same system as used previously that been further developed, this could be referenced once in the Methods. If not, I would expect an earlier reference, or perhaps no reference at all for this.

35 does not seem the best reference for high laser intensities sending molecules into the dark state.

36 is incomplete.

Method:

- How can the read find the same galvanometer mirrors?
- Key distances along the optical path in Fig. 1 should be described somewhere..
- The magnification of the flat-top profile appears to be done by L3 and 4, not L2 and L3 (Fig. 1

legend).

- The use of Michelson contrast seems strange (and probably needs defining somewhere). Why not use the intensity of the field at different positions, which is intuitively simpler corresponds more clearly to variation in fluorescence data obtained?
- What control software was used for the whole system, including the galvanometer mirrors?
- How, specifically, is the system modified to change the illumination profile, including changing the uniform field width (currently "by adapting the scanning path") and changing from ASTER to Gaussian modes?
- What was the exposure time for Gaussian profile STORM imaging?
- Please explain or at least reference Feret's diameter.
- Hollowness was found to be "rather independent of the size of the cluster". Data and statistics should be provided for this.

Other specific comments on figures:

- Fig. 2: The X and Y-shaped are not easily distinguishable or legible.
- Suppl. Fig. 3: The Y markers are hard to see. Panels f and g do not appear in the figure legend.
- Fig. 3: The highlights for the selected areas are very hard to make out.
- Fig. 4: A scale bar for panel b would be helpful.
- Fig. 5: Blue and green indicators are hard to distinguish in this figure. Can some scale information be provided for the Fourier transforms? Are they autocorrelations? How were they produced (for Methods section)?

I did not see a reference to Suppl. Fig. 4 in the main text.

I recommend that these comments are addressed before considering the publication of this paper.

Detailed responses to reviewers

Revised version manuscript NCOMMS-20-18746

We would like to thank the two reviewers for their thorough assessment of our manuscript and their extremely useful comments. We believe that by addressing these comments the manuscript has drastically improved. Major modifications revolve around temporal effects of ASTER and its compatibility with both live cells and single molecule imaging. We have overhauled Figure 1 to illustrate the time averaging effect, simplified the presentation of field uniformity, extensively modified the text, and added 7 supplementary figures and three supplementary notes. Minor modifications have also been made to all figures to improve clarity. We have thoughtfully addressed each concern of the reviewers and provide below detailed responses (in blue) for each. All changes within the main manuscript and supplementary materials are highlighted in this revised version.

Reviewer #1 (Remarks to the Author):

In this manuscript Adrien Mau et al introduce a new illumination method, called ASTER, for achieving SMLM with large field of view. The authors claim that ASTER is capable of providing uniform illumination without suffering the limitations from reported methods (e.g. laser damage to objective lens, power loss, or bleaching the whole sample at once). The method reported in this manuscript seems promising, **but the current evaluation is insufficient**, and the manuscript is loosely written. **I cannot determine the performance of the method relative to others.**

My major concern is that, since the ASTER method is based on 2D scanning of a Gaussian beam, the total time required to scan the full field of view (FOV) would set a hard limit for the minimum camera integration time, which has a direct impact on the temporal resolution of STORM imaging. Actually, the camera integration time used in the manuscript is typically 50-100 ms, which is OK for DNA-PAINT, but may be too long for conventional STORM experiments (where researchers would like to use 10 - 20 ms to increase the temporal resolution). The relatively long camera integration time would surely limit the spread of the ASTER method in a wider application field. The authors may consider a multi-beam strategy to break this limitation. Anyway, more works on pushing the minimal camera integration time should be done to justify a publication in a high impact journal, such as Nature Communications.

We thank the reviewer for this remark as indeed the timing is an important parameter for both live cell and single molecule imaging. We have thus introduced new data to clearly demonstrate the temporal characteristics of ASTER and assess its benefits to both imaging modalities.

The fundamental limit is given by the scanning speed of the galvanometric mirrors, which is 3.3 kHz with the current model used. Depending on the size of the input gaussian beam and of the observed FOV, the input beam will be scanned over an optimal number of lines to generate a uniform excitation (cf. Supplementary Fig. 1). For 200 μm x 200 μm with an input gaussian beam size of $\sigma = 17 \mu\text{m}$ scanned over ten lines, 3 ms is the limit to generate the flat-top illumination. We use the galvanometric scanners at slightly lower speed, and the fastest scanning time used for 200 μm x 200 μm is typically 5 ms. Fig. 1 has been modified to include a schematic of the time averaging during the scanning time. A new Supplementary Fig. 2, now illustrate the fastest scanning time used (5 ms) and other scanning time options. For acquisition times longer than the scanning time, two parameters can be adjusted: lowering down the scanning speed or/and repeating the scanning several time by keeping

the initial speed. The ideal, however, is to optimize by playing with both. Following this question, we have also introduced in Supplementary figure 4 an image in classical TIRF where we set an integration time of 5ms equal to the scanning period of 5ms, and imaged over the whole (220 μm x 220 μm) FOV of the camera. However here the limitation is the frame rate, as for such a wide field the camera can't transfer faster than 10 ms. This nevertheless shows that we can scan the entire FOV at 200 fps and confirms that dynamic samples could be observed under ASTER excitation.

This scanning time is compatible with STORM acquisition of 10-20 ms, and with the frame rate of our camera of 100 fps associated to this large field of view. As discussed in the next answer, we investigated the impact of the scanning time on STORM acquisition, which can be increased to match the acquisition time or kept with its nominal speed to average over multiple scans. As represented in the new supplementary figure 10, we observed that both strategies result in the same blinking kinetics. This finding is now described on lines 214-216 of the main text. The STORM images presented in the manuscript were acquired with a scanning time of 25ms, and an integration time of 50 ms which is optimal for the dye used (AF647) in conjunction with the used buffer, i.e. most molecules are seen in only one image. This of course could be scaled down towards the minimal scanning time in conjunction with appropriate dyes and buffer. The acquisition parameters have been added in the legends and methods.

Alternative acquisition strategies have also been proposed in STORM (Lin et al, PLOS one 2015), where very high intensity is used in conjunction with fast acquisition to allow one to observe multiple region of interest associated to smaller field of view without compromising the precision. The faster the camera speed (down to 1600 fps), the higher the intensity (16–97 kW/cm²) is required to achieve optimal localization precision. The flexibility of ASTER allows to easily implement and optimise this acquisition strategy, as both the input gaussian beam and field of view size can be adjusted to obtain the optimal high irradiance. In Supplementary Figure 12 we now present a STORM image acquired in 100 seconds with an irradiance of 27 KW/cm².

Altogether, these new data confirm that ASTER offers an original approach to observe single molecules with optimal and uniform irradiance, by allowing adjustment of multiple parameters, and within adequate acquisition times that can leverage the framerate of current sCMOS cameras over large fields of view.

Other concerns or comments:

Fig 1: The authors mention a temporally averaged flat-top excitation profile. Any further characterization on the time-dependent behavior of the illumination?

As the reviewer highlights, ASTER excitation is at each given instant gaussian, and this may have effect on the behaviour of fluorophores as they do not receive a constant signal but a fast varying irradiance. In PAINT and STORM, the fact that fluorophores yielded similar blinking, independently of their position on the scanning path indicates that the determining factor is the total number of photon delivered, and not the temporal distribution of the excitation. This observation has been completed by further experiments, presented in Supplementary Figure 10. Fluorophores were excited in STORM at different scanning speeds on the same field, so that the same number of photons was always provided during a single 50 ms frame but in episodes of varying length within that frame. As the reviewer will notice, the blinking statistics remained relatively unchanged for all scanning speeds: the number of photons, blinking ON-time and density of fluorophores do not exhibit significant variation. Zoomed areas in photon count and ON-time distributions show that the flux of fluorophore emission

decreased over time. This is in fact not due to the different scanning speeds, as a final scanning period of 50ms shows different statistics than the first 50ms period scan, but to a temporal variation presumably highlighting consumption of the buffer. References to these new data have been introduced within the main text in line 213-215.

Fig 1d shows a slightly rotated square, and Fig 1e shows slopes in the illumination profile. Both of them are not desired for further image stitching. What are the origin of these and any ways to get rid of them?

The rotated square is linked to the axis along which galvanometers are scanning, compared to the camera. This can be solved by a re-alignment of the galvanometers, by adapting the input scanning path command, or by a rotation of the camera rotation, which could be constrained in some setups. To demonstrate that this can be solved, galvanometers have been re-aligned and a new image has been introduced in Fig 1e, where the homogeneous image of Nile Blue leads to a straight square compared to FOV border.

The gaussian shaped borders arise because of the intrinsic scanning of a gaussian beam. Using a smaller gaussian beam will sharpen these borders, with the side-effect of increasing scanning time as more lines will be needed to synthesize a flat illumination profile. This point has been described more in details in lines 92-94, along with comments on Nile Blue profile figures represented on fig. 1c.

Global slope in the illumination profile is also due to vignetting, which is now clearly addressed, at lines 94-100 and 210-212. Supplementary Fig.3 shows the impact of vignetting on field uniformity.

To optimize stitching, slightly overlapping FOVs are acquired, which eases FOV association but also permits to crop the border of the FOV that is not uniform.

Fig. 2. The authors demonstrate epifluorescence and TIRF imaging of a large FOV of 200 μm x 200 μm , but provide no details on how to enable such a large FOV. And, what is the relationship between the minimum camera integration time and the corresponding FOV? Fig 2d-e should be analyzed quantitatively, rather than presenting only those pretty images.

We achieve wide FOV excitation by increasing the amplitude input for scanning, while assuring a flat-top profile by scanning a sufficient number of lines. We now detail how we switch FOV size (line 52-53, line 89) and underline more clearly the condition on flat-top emergence at different field size. We have also added more details on the relationship between FOV size and minimum integration time in Supplementary note 2, and conclude on the temporal limit of our implementation.

Concerning fig. 2d-e, we have added intensity cross profiles to further assess the uniformity of the EPI and TIRF images.

Fig 3: What method was used to calculate the localization precision?

For each individual nanoruler, our goal is to estimate the parameters of three normal distributions that correspond to the three spots. A Gaussian Mixture Model (GMM) fits the 2D distribution of point cloud with three normal distribution sources, and estimate each of their standard deviation and center. For each nanoruler, the localisation precision is then measured as the mean of the three standard deviations. Notably, this measure will take molecule blinking density in account and may be

comparable to the experimental resolution. This is now more clearly explained in Methods and Supplementary Figure 8.

Fig 4: It would be more convincing to include a highlighted area in the corner of the image.

We now included the far-left corner of the image in the highlighted areas. There we could measure a slight decrease in photon count but no drastic modification of ON-time distribution and evolution of detections along frames. As only photon count was impacted, this suggests that excitation is similar to the other parts of the field, but inhomogeneity arises from imperfect detection. We believe this is due to vignetting, it is a classical phenomenon in photography and optics and is not surprising with the use of our objective. Indeed, work from Kurvits et al. mention vignetting on wide field of view for a similar Nikon 1.49 x60 objective: there is a decreasing transmission at 1.2 numerical aperture that is even more drastic for molecules close to the coverslip as most of the supercritical-angle fluorescence signal is lost.

As the purpose of Fig.4 is to highlight that uniform STORM blinking stems from uniform excitation, including one of these corner may mislead the readers but we are now commenting this effect in the article as it still is an inner limitation for wide homogeneous imaging and must be brought to light. Presumably, this has low impact on excitation as we do not cover the whole BFP but this impacts detection. This can also be noticed in other works: Zhao et al. Fig.9-c has lower photon count for areas close to the corner, and higher photon count at the center, even though their illumination is uniform.

To raise more awareness about this issue, we added mention to vignetting when commenting Nile Blue profiles and the STORM image from Fig.4. We also excited a thin layer of Nile Blue with an homogeneous $220\ \mu\text{m} \times 220\ \mu\text{m}$ field and show that vignetting becomes significant at a radial distance $r=115\ \mu\text{m}$ from the center (Supplementary Fig. 3). Thus, fields of $220 \times 220\ \mu\text{m}^2$, which reach a radius of $155\ \mu\text{m}$ at their border can hardly be detected in a uniform manner, an effect that hampers all available objectives. $220\ \mu\text{m} \times 220\ \mu\text{m}$ images can still be obtained as presented on supplementary 6, but we rather preferred to present $200\ \mu\text{m} \times 200\ \mu\text{m}$ images where the impact of vignetting is limited to a 20% loss of intensity at the borders (affecting 9% of the field). As also mention in the text (lines 102-103), ASTER can also offer a scanning following an Archimedes spiral (supplementary 1), which offers for a radius of $113\ \mu\text{m}$ an equivalent area of $200 \times 200\ \mu\text{m}^2$ free of vignetting (Supplementary Fig. 3).

Page 9, the last paragraph. The authors claim that they achieve the largest FOV of $200\ \mu\text{m} \times 200\ \mu\text{m}$. Actually, an even larger FOV ($221\ \mu\text{m} \times 221\ \mu\text{m}$) was reported in reference 24.

We apologize if our paragraph was misleading, we demonstrated here a $200\ \mu\text{m} \times 200\ \mu\text{m}$ and we didn't want to claim for the largest field of view ever achieved, as we believe ASTER benefits not only arises from the large FOV of view, but also through its flexibility in terms of optical sectioning, irradiance, acquisition speed... We agree with the reviewer than reference 24 achieved a larger field of view which cover the whole chip of the SCMOS camera $221 \times 221\ \mu\text{m}^2$ FOV). However as clearly shown in reference 24 (fig. 10a), taking advantage of the whole chip is limited by the microscope mount and optics itself and corners are clearly not well illuminated. Regarding the comparison with reference 24, as mentioned in the introduction, this approach relies on a high power laser (initially 6W) coupled into a multimode fiber which delivered more than 2W. As mentioned by the authors, this prevents the application in oblique angle or TIRF illumination in a through-objective configuration, the most commonly used configuration for SMLM experiments. These comments are now synthesized at lines 219-222 of the main text.

We now have also included supplementary figure 3, which illustrates a similar vignetting issue as discussed in reference 24.

Page 10, the first paragraph. The calculated irradiance of 0.5 kW/cm² may be misleading. Since this is basically a point scanning technique, it is better to report the intensity of the Gaussian beam on the sample plane.

We now report both instantaneous irradiance of the input gaussian beam, and mean irradiance on an integrated camera frame. This will indeed be easier for readers to then grasp why ASTER efficiently sends molecules in the dark state even though the global irradiance is low.

Page 12. The integration time in TIRF imaging was 200 ms. Why did the authors use such a long integration time?

For such classical wide-field experiment, 200 ms integration time was used to increase signal to noise ratio. In this matter, we were not limited by ASTER: the scanning period used for this experiment was 50 ms, so that one image corresponds to four successive scans of the whole image flat-top profile. It is also possible to image in TIRF at 5ms integration time, as shown in Supplementary Figure 4. We hope that this will help the reader understand that ASTER in TIRF is not limited to imaging fixed sample, but may also be used for imaging live and fast dynamical events.

Supplementary Fig 7. It would be good to see statics similar to those shown in Fig 4d-f.

Statistics similar to those of Fig. 4 were added. Interestingly, while for each image similar areas exhibit similar blinking statistics, a temporal effect is present and statistics slightly differs from one image to another. This is most probably due to buffer consumption as our STORM sample was not sealed. This highlights that even though quantitative analysis can be performed on each individual image, samples needs to be correctly protected from oxygen or a correction factor will be needed in order to efficiently quantify the whole stitched image. Had it happened under gaussian illumination, both a spatial and image by image correction would be necessary.

It is confusing to see the different FOVs in the SMLM images: 120 um x 120 um in Fig 3c, 200 um x 200 um in Fig 4, 140 um x 140 um in Fig 5a, 200 um x 200 um in Fig 5d, etc. Why did this happen?

We apologize for any confusion the variety of field sizes shown may have caused. Smaller fields of view were observed in order to offer a fair comparison between the FOV usually observed with the conventional gaussian field excitation (Fig. 3). All images acquired in conventional dSTORM, and not intended as a direct comparison with a gaussian excitation, were initially designed as 200 μm x 200 μm . However, for the clathrin image (Fig. 5a), we balanced speed of blinking and field of view size to optimize the experiment. The 140 x 140 μm^2 field is already 20 times wider than the classical 30 x 30 μm^2 and yielded a large number of clathrin structures for statistical analysis. All of this is now clearly detailed in Methods.

The authors present only 2D imaging results. It would be nice to see the performance (at least discussions) of this ASTER method in 3D STORM.

We thank the reviewer for this suggestion, indeed ASTER is directly compatible with 3D imaging. In addition to discussion about combining ASTER and 3D detection methods, we added Supplementary

Figure 14, which presents a wide-field 3D image of microtubules performed by inducing astigmatism via a cylindrical lens.

The authors should revise the manuscript carefully to enable a brief and precise writing. For example, in the last paragraph in page 9, the term “SMLM PAINT” looks odd, and the last sentence should be deleted.

We apologise for mistakes and typos that evaded our careful proofreading, and the manuscript has been revised to correct this particular point.

Reviewer #2 (Remarks to the Author):

This paper claims the novel development of scanned widefield illumination for use in epifluorescence, HiLo and TIRF microscopy. The scanned widefield illumination is claimed to produce a larger, more uniform FOV, using lower laser powers, than has otherwise been possible.

However, the method of scanning a wide illumination beam, either for transmitted light or fluorescence microscopy is already known as adaptive scanning [Benjamin Potsaid, Fern P. Finger, John Ting-Yung Wen, "Living organism imaging with the adaptive scanning optical microscope (ASOM)," Proc. SPIE 6441, Imaging, Manipulation, and Analysis of Biomolecules, Cells, and Tissues V, 64411D (19 February 2007); I would expect this to be referenced, and perhaps the Optics Express 2005 paper preceding it, which presented adaptive scanning but did not include fluorescence. Therefore some of the claims on novelty would need to be removed or modified.

There are also TIRF illumination scanning schemes used for structured illumination microscopy (TIRF-SIM) and single molecule localization microscopy (SMLM) (e.g. SIMFLUX, SIMPLE), and the editors may consider it relevant to present the novelty and advantages of the authors' approach in reference to these. These existing approaches involve scanning and are more complicated than the authors' approach, but in them scanning is over smaller distances using different methods and does not address the same problems.

As the reviewer highlights, scanning is a powerful tool that can be used in a number of ways. The mentioned publications indeed focus on improving detection: By using a high speed scanning mirror in a post-objective configuration, the ASOM captures a complete image at each scan position and assembles image mosaics on the fly without moving the sample, so mainly intend to enhance the detection. On contrary, ASTER focuses on the control of the excitation, both in terms of uniformity and adaptability, which represents a very active field in particular for the single molecule imaging community. One application of ASTER is the possibility to extend several fields of view by stitching. However, the main goal of ASTER is to provide a controlled and uniform irradiance, key parameters to achieve uniform precision in SMLM. As mentioned within the introduction and new supplementary notes, despite multiple strategies recently proposed for SMLM, ASTER reaches an appealing balance regarding laser power, optical sectioning and adaptability.

In fact, ASTER could even be combined with the ASOM to perform wide field acquisitions without having to move the sample, in particular this could be a key combination for Single Particle Tracking. The potential of this combination has been added in the conclusion perspective paragraph

Regarding the use of scanning for TIRF/SIM, we mainly found references (Chen et Al. 10.1117/1.JBO.23.4.046007, Roth et al. 10.1364/BOE.391561 ..), were piezo or galvanometric mirrors

are used to switch between multiple optical paths associated to multiple grating positions necessary for SIM imaging. In this configuration, they are similar to a spinning azimuthal TIRF configuration, as they shift position of the beam in the Back Focal Plane but maintain a fixed position of the beam at sample plane. While ASTER presents the opposite conjugation: the beam is fixed in the back focal plane and shifted at the sample plane.

Concerning the mentioned SIMFLUX and SIMPLE, they used a structured excitation generated through physical grating or generated through a DMD, but none of them uses a scanning device, or any mean to provide a uniform precision. The goal of SIMFLUX/SIMPLE is to provide an enhanced precision through the structured excitation, presented in restricted field of view ($25 \times 25 \mu\text{m}^2$), as they are based on gaussian excitation any larger field of view will suffer from localization which varies across the field of view. As no uniform excitation for these kind of approach in localization microscopy has been proposed so far, we have added this prospective into the conclusion along with the papers on this topic.

The application of adaptive scanning for wide and uniform fluorescence excitation in TIRF and SMLM may be considered novel and will be of interest to others in the field. The paper presents promising results on precision uniformity in SMLM and large FOV and stitched SMLM images. It may lead to flexible, higher throughput data acquisition with less specialized/expensive equipment in TIRF-SMLM, which would improve or accelerate experimental results. Further, quantitative comparison with the flat-field epi-fluorescence SMLM schemes of Douglass et al. (already referenced) and piSMLM would be useful to explain any advantages in epi-fluorescence.

Following this remark, we have included an in-depth comparison between existing setups and ASTER in Supplementary Note 1, detailing the drawbacks and benefits of each individual solution.

This work may also lead to acquisition of data from larger samples without requiring image stitching, or requiring less image stitching. Particularly, if the claims of equivalent localization data quality but with lower laser power are robust, labs that require large FOV/high-throughput SMLM data may be interested in applying this technique themselves.

An example of HiLo imaging with the larger or more uniform FOV would be useful for illustrating the claim that the method is useful here.

In fact, Fig. 4 excitation was already performed in HiLo but we failed to communicate this point clearly. Methods were only mentioning an “oblique illumination” for STORM which has now been changed by the more precise term of HiLo. Indeed, we used this configuration to make the best out of available laser power.

Discussion of the effect of modulating the excitation of single molecules would be welcome: In ASTER SMLM, are potential photons not acquired because scanning the illumination moves away from an emitting molecule before it has finished emitting?

Discussion and Supplementary Figure 10 have been added to emphasize the effect of modulating intensity. This issue has also been highlighted by reviewer#1. As the reviewers noticed, because of ASTER’s modulation, the fluorophore will not receive the same irradiance at each moment, and during one scan period, it may not receive enough excitation photon to finish its emitting cycle. Concerning modulation of irradiance, we show that this does not affect fluorophore blinking significantly:

fluorophores in Supplementary Figure 10 yielded similar blinking statistics as long as a similar number of photon was provided to them over a given integration time, independently of irradiance episodes within the integration time.

Similarly, to classical STORM experiments, emitting molecules may be spotted on consecutive frames when they do not receive enough photon (i.e do not perform enough cycles) between the time of their return to the ON emitting state and the end of the camera integration frame. For ASTER STORM experiments we noticed that molecules mostly emit all photons on one frame (as can be seen in Fig4.d), with ON-time distributions similar to those of continuously illuminated STORM experiments. Under classical continuous illumination, a molecule will likely remain on successive frames when it goes back to an ON-state close to the end of the frame integration.

With a fast scanning period, such as 5ms scanning and 50 ms integration time, molecules will behave in a similar manner than that of continuous illumination as the flat top is averaged 10 times during one frame acquisition. However, if the scanning period is the same as the integration time each line of the scanning path is only scanned once during a camera integration frames and effects such as those mentioned by the reviewer may happen: a molecule that reach an ON state on a part of the field that has already been scanned will likely receive few photons and appear on the next frame. As can be noticed on Supplementary Figure 10, scanning speed did not impact the density of active molecule, or the mean ON time of molecules, which indicates that the same number of molecules remain on consecutive frames, independently of the scanning path.

It should be noted that in STORM, high irradiances speed up the blinking process in a nearly linear manner, so that even though the time during which the fluorophore is excited is in the order of a few ms, this is sufficient for the fluorophore to perform a full blinking cycle and reach a long-lived off state.

However, more rigorous presentation of results is required in the text and supplementary material in order to justify the claims of these developments for TIRF and SMLM. For instance:
- The choice of $1.2 \cdot \sigma$ for the upper limit on the distance between scan lines is not illustrated or derived, including in Suppl. Fig. 1.

The choice of $1.2 \cdot \sigma$ as an upper limit is now included in supplementary figure 1 and also detailed in Supplementary Note 2. As represented and discussed a larger gap between 1.2 up to 1.7 can be chosen while preserving uniformity.

- Confidence intervals on results are needed in the main text in many places.

Confidence intervals were added for measurement of TIRF penetration depth and nanoruler localisation precision.

- Detail of the calculation of the TIRF penetration depth in supplementary would be very useful.

We now added Supplementary Figure 6 and Supplementary Note 3 to detail all parts related to TIRF penetration depth analysis and calculation.

- Quantitative results for spatial correlation and anisotropy are required when these are mentioned for the TIRF field.

Quantitative results are now available concerning global and local spatial correlation of measurements. This can be found in Supplementary Figure 5 and Supplementary Note 3.

- A limit on measurement precision is mentioned as explaining variability in results per bead when measuring the TIRF field. What is the source of this and what is its magnitude?

We have noticed since Ref 33. (Aberration-accounting calibration for 3D single-molecule localization microscopy) that some beads tend to flatten on the surface of the coverslip while other remain relatively spherical, which could be the first source of variability. However other issues may also be relevant, such as variation in the coverslip, local index mismatch...

It is not straightforward to estimate the magnitude of this phenomenon, we estimated the precision of our measurement in Supplementary Note 3 and conclude that this effect impacts measurements within the same order of magnitude than that of our estimated precision.

- More precise comparison of the images in Fig. 2d,e is needed. Fig. 2d,e may currently be mislabelled in the figure legend.

Cross profiles are now added to the TIRF figure to better highlight the sectioning and efficiency of the technique. We apologize for the mislabeling, the legend has been corrected.

- Showing the laser speckle/interference that ASTER helps to deal with would be informative in supplementary material.- It is claimed that the TIRF images obtained are comparable with a spinning azimuthal TIRF system, but no evidence is provided for this.

Indeed, we noticed on various TIRF experiments that no interference pattern was noticeable, though we did not quantify this or compared it to a scanless experiment. Following, this suggestion we now have introduced to strengthen this claim an experimental proof, we modified ASTER's setup so that one path of the experiment could perform azimuthal scanning. We then compared on biological sample the resulting TIRF images without scanning, with azimuthal scanning and with ASTER scanning (Supplementary Figure 7). Even though it will still be subject to shadowing effects, ASTER yield similar performance to azimuthal spinning, with an appreciable extinction of interference patterns and the advantage of providing an adaptable non-gaussian excitation profile. This is now commented at lines 133-135 of the main text.

- What are the sizes of the Gaussian beams used for Gaussian and ASTER illumination in the different fields in Fig. 3?

For ASTER illumination Fig.3 the same beam size was used than the one from Fig.1 and 2: $\sigma = 17 \mu\text{m}$. When comparing with a continuous gaussian illumination, a constant beam size of $\sigma = 45 \mu\text{m}$ was used. This is indicated within the Methods.

- Quantitative discussion of the Fig. 3 results needs attention in the text. The uniformity of the ASTER results comes across as over-represented: precision is not uniformly 7.6 nm or 9 nm according to Fig. 2g. The range of mean values over the different positions, or confidence intervals would be much more useful.

We modified the claim of uniform precision by indicating mean value and confidence interval for ASTER results, along with a range of values at different parts of the field.

The non-uniformity of the Gaussian results also comes across as over-represented in discussion of Fig. 2h, where the size error is fairly uniform up to about 30 μm out from the centre. There is obviously a real difference between the scheme, but it does not seem to have been presented correctly in the text.

Performance of the Gaussian illumination is now more fairly described in term of uniformity.

- It is curious that the Gaussian distribution had a smaller mean precision at the centre of the FOV, but a larger ruler size error. Why might that be?

This is indeed curious and rather counter-intuitive. This would underline that even though the precision is better, the confidence is not. AS DNA-PAINT is also prone to bleaching, higher irradiance at the centre can also reduce the total number of detected events, reducing also the statistic.

High irradiance could preferentially enhance noise compared to the number of emitted photons but this would hamper precision and not confidence. It should also be noted that measurement being taken on constant radial intervals, it contains few nanorulers at the centre and thus has high imprecision. Consequently, we do not think a confident conclusion can be made on this matter.

- Suppl. Fig. 6 contains low quality images that are hard to read and therefore hard to interpret in places. Presenting a statistic like median photon counts to go with panel c might be informative. Panel g is not explained.

Image of now Supplementary Figure 9 has been edited to be of higher quality and presents an intuitive radial distribution of median photon count. The two last panels are now explained.

What was the Gaussian beam size used in ASTER for Fig. 5, and what is the inter-line spacing? (My guess is 20 μm spacing, but I am not sure.)

The gaussian beam size used for ASTER is $\sigma = 17 \mu\text{m}$ by default. For wide 200 μm x 200 μm fields a spacing of 24 μm has been used (gap of 1.4σ). A 20 μm spacing is used for smaller fields (gap of 1.2σ). Information has been added within the Methods section.

- Where is the number $\sim 1,500$ clathrin-coated pits obtained from for a classical 30 μm x 30 μm FOV?

This number is estimated via a cross product with FOV sizes. As we found 20000 clathrin cluster on 200 μm x 200 μm , we should expect 1500 on 30 μm x 30 μm . (To be exactly rigorous, a 30 μm x 30 μm FOV could be cherry-picked on an area with many clathrin clusters, while on a wide FOV we will ultimately always image some void areas, so that a wisely selected 30x30 μm^2 FOV could maybe image up to 2000-2500 clathrin-coated pits.)

- The holes suggesting 18 nm side-length fenestrations are not currently clear in Fig. 5. Does the estimated localization precision achieved suggest that these would be resolvable in images? More information on scale in b and c is needed and some higher-magnification images may help. Following this suggestion, images of Fig. 5 have been magnified to better present details of clathrin coated pits. With our estimated localisation precision, holes should be challenging to resolve, but not impossible. Depending on stochastic events of blinking, some fenestration holes may appear closed due to detection of overlapping fluorophores while others will remain resolvable. Scale has now been added and images have been magnified.

Paragraph 1: “uniform excitation over a tunable FOV” would be sufficient and better: every field has 95% uniformity out to a certain distance, even if it smaller. “has advantages over” in these areas would be better than “outperforms”, since few quantitative performance comparisons are made.

This has been changed accordingly

Paragraph 2 and 3 seem unnecessary as quite a long repetition of previous results (in which the localization precision claimed in the ASTER fields is given as even lower, 7 nm, than in the results section, 7.6 nm, which also needed addressing).

We have modified the text, and the first part of the conclusion has been reduced. The performances are now more quickly reminded but we are ready to further reduced if needed.

Paragraph 4: Robust reliability is claimed, which is not evidenced (and sounds a little tautologous). The particular connection between ASTER and ref. 44 is not clear. I would say “ASTER may be used in combination with improved detection schemes [45, 46]”, rather than that they are further uses of ASTER. The relevance of flat-fielding to photolithography should at least be referenced.

We have followed the suggestion, and modified the text and added references concerning the photolithography.

Citations: 31 and 32 are surprising here. If it is the same system as used previously that been further developed, this could be referenced once in the Methods. If not, I would expect an earlier reference, or perhaps no reference at all for this. 35 does not seem the best reference for high laser intensities sending molecules into the dark state. 36 is incomplete.

The citations 31/32 where here as earlier implementation of the use of a translation stage to precisely position the beam in the back focal plan to switch with a high reliability from epifluorescence excitation to TIRF. We deleted citations 31 and placed citation 32 in Methods. Citations 34 and 35 have been updated

Method:

- How can the read find the same galvanometer mirrors?

Mirrors are 6125H from Cambridge and are now mentioned in methods. We apologize for this unfortunate oversight.

- Key distances along the optical path in Fig. 1 should be described somewhere..

Distances along the optical path are always chosen as the sum of lenses, this guarantee a telecentric excitation system and has now been added to the methods.

- The magnification of the flat-top profile appears to be done by L3 and 4, not L2 and L3 (Fig. 1 legend).

This has now been corrected.

- The use of Michelson contrast seems strange (and probably needs defining somewhere). Why not use the intensity of the field at different positions, which is intuitively simpler corresponds more clearly to variation in fluorescence data obtained?

Reference to Michelson contrast has now been deleted for more intuitive comparison on intensity.

- What control software was used for the whole system, including the galvanometer mirrors?

Neo-Live (Abbelight) was used and are now mentioned in Methods, along with the galvanometer input device.

- How, specifically, is the system modified to change the illumination profile, including changing the uniform field width (currently “by adapting the scanning path”) and changing from ASTER to Gaussian modes?

We now mention how switching from one ASTER field size to another can be done by simply decreasing or increasing input amplitude (though conditions on the gap between lines must be met). To switch to a gaussian illumination, one may set ASTER scanning to a simple offset and then magnify the base gaussian beam by physically adding an afocal system. This is now clearly described at lines line 52-53 and 89 of the main text for adaptation of field size, and in Methods for switching from ASTER to Gaussian.

- What was the exposure time for Gaussian profile STORM imaging?

It was 50 ms. This is now detailed in Supplementary and Methods.

- Please explain or at least reference Feret’s diameter.

A reference to Feret’s diameter has now been provided. Feret’s diameter is the largest measurable distance that can be measured on a caliper holding the object, and is a common occurrence in measuring and comparing particles, though it is also referred as ‘caliper diameter’.

- Hollowness was found to be “rather independent of the size of the cluster”. Data and statistics should be provided for this.

Hollowness was indeed found to be rather independent from the size of cluster, for cluster sizes of 30 nm and beyond. However as this information doesn’t serve any narrative in the manuscript we would preferably remove it from the methods. Another solution would be to add this image in the Supplementary.

Other specific comments on figures :

- Fig. 2: The X and Y-shaped are not easily distinguishable or legible.

We now use cross and circles which are more distinguishable.

- Suppl. Fig. 3: The Y markers are hard to see. Panels f and g do not appear in the figure legend.

We now also use cross and circles which are more distinguishable.

- Fig. 3: The highlights for the selected areas are very hard to make out.

Indeed, it was first optimised to not hinder the appreciation of the scatter plot but in the end lost its intended purpose by not being distinguishable enough. We now use a more opaque border and reached a better compromise. As for clathrin-coated pits, nanoruler images were magnified so that readers could more efficiently identify key aspects of the figure.

- Fig. 4: A scale bar for panel b would be helpful.

A scale bar has been added in panel b

- Fig. 5: Blue and green indicators are hard to distinguish in this figure. Can some scale information be provided for the Fourier transforms? Are they autocorrelations? How were they produced (for Methods section)?

We now magnified the highlighted region to increase the visible area of green and blue populations. We added scales on Fourier transforms and removed text next to scalebars as it was not consistent with the designs of other figures.

Fourier transforms are basic 2D-Fast Fourier Transform performed with the `fft2` function from the Python `numpy.fft` library, as is now detailed in Methods. Autocorrelations may also be used highlights the 190nm frequency.

I did not see a reference to Suppl. Fig. 4 in the main text.

Indeed we apologize, there was no reference to it. This figure had its purpose in more intuitive understanding of TIRF as the disappearance of nucleus efficiency highlights that parts far from the coverslip are not excited anymore. With the added 5ms TIRF image of microtubules it is now rather redundant and has been deleted.

I recommend that these comments are addressed before considering the publication of this paper.

We would like to thank the reviewer again for her/his constructive remarks, and we hope that she/he will be satisfied with the various changes made in response to all these suggestions.

REVIEWER COMMENTS

Reviewer #1 (Remarks to the Author):

The revised manuscript is significantly improved, and could be published after correcting the typos and Grammar mistakes and improving the readability. A few examples: 1) Line 24, the “f” before “where” looks strange. 2) Line 67-68, Tint and Tscan are better to be italics. 3) Line 223: ref 24 should be 27.

From: Prof. Zhen-li Huang, Hainan University, China

Reviewer #2 (Remarks to the Author):

The authors have made several helpful changes and significantly improved the manuscript. The technique now seems generally validated as the claimed advance in imaging technology, with some interesting technical notes relevant to the field as part of that validation.

However, further, more minor, edits should be made before publication:

- Fig. 5:

-- The ‘holes’ that are claimed may be 18 nm fenestrations are not yet clear in the figure. These should be highlighted in the images. If they cannot be convincingly pointed out, the discussion (lines 248-250) should be modified.

- Confidence intervals:

-- Towards the start of the paragraph starting on line 157, a similar mean \pm confidence interval for localization precision with the Gaussian beam would also be informative for comparison.

Presumably, the confidence interval would be considerably larger for the Gaussian beam than with ASTER.

-- These confidence intervals need to be explicitly defined at some point (e.g. as mean \pm s.d.)

- Ref. 48 is incomplete.

- Suppl. Note 1:

-- Include references in the text and table.

-- Explain why 16 lines is a useful number for ASTER in this comparison (e.g. the size the resulting FOV, using a Gaussian beam with $\sigma = x \text{ nm}$, and a line spacing of $x \sigma$).

- Suppl. Note 2:

-- Define n_{minline} and T_{min} explicitly, for clarity.

-- 'scans lines' should be changed for clarity, maybe to 'line scans'.

-- Why does D need to be the longest length – why should it not be the shortest length? It would be good to explain this, or just settle for 'the length dimension D ', rather than 'its longest length'.

- Line 137:

-- 'spinning TIRF would prevent further quantification': This should be edited to specifically refer to the larger FOV allowed by ASTER, rather than implying more general problems with spinning TIRF.

- Suppl. Note 3:

-- Line 2 and 5: Should this refer to Suppl. Fig. 5, not Suppl. Fig. 3?

-- Line 1: The mean TIRF penetration depth is 117 nm in Suppl. Fig. 5 and the main text. Should this replace the 115 nm currently given in Suppl. Note 3? If not, where does it come from?

-- The sentences discussing the precision limit due to pixel size should be condensed for clarity.

- Suppl. Fig. 7

-- There seems to be no panel (b). It still makes sense, and it is up to you and the editors whether that needs changing.

-- It is not currently obvious that laser speckle is specifically seen in (c), although this is implied by 'interference speckles' in main text line 134. Either the presence of laser speckle should be discussed in Suppl. Fig. 7 if the authors wish to highlight it in Suppl. Fig. 7c, or more generally, the main text could be changed to 'interference patterns.'

- Ref. 34 is incomplete.

- The statement that hollowness is 'rather independent of the size of the cluster' in Methods.

-- The authors suggested two ways to edit this in their response to my comment. I recommend the option of removing that statement/information, unless it is to be given in the main text and justified

in new supplementary information. Use of the image sent with the authors' response may introduce further questions if used as is.

- Fig. 5:

-- In the legend for (d), 'inlet' should read 'inset'.

-- The line profile in (e): apart from the shape of the line, details of that plot are poorly legible.

- Line 287:

-- 'Better resolution' should instead refer to localization precision, concerning the result for the 70 μm FOV.

Detailed responses to reviewers

Revised version manuscript NCOMMS-20-18746

We would like to thank the two reviewers for their thorough assessment of our revised manuscript and their extremely useful comments. We have thoughtfully addressed each concern of the reviewers and provide below detailed responses (in blue) for each. In particular, a new supplementary figure 5 has been added, along with suggested text and figures modifications. All changes within the main manuscript and supplementary materials are indicated in this response.

REVIEWER COMMENTS

Reviewer #1 (Remarks to the Author):

The revised manuscript is significantly improved, and could be published after correcting the typos and Grammar mistakes and improving the readability. A few examples: 1) Line 24, the “f” before “where” looks strange. 2) Line 67-68, Tint and Tscan are better to be italics. 3) Line 223: ref 24 should be 27.

From: Prof. Zhen-li Huang, Hainan University, China

We thank Pr Zhen-li Huang for his positive feedback on the revised version, and for the typos changes that have been addressed in the new revised version.

Reviewer #2 (Remarks to the Author):

The authors have made several helpful changes and significantly improved the manuscript. The technique now seems generally validated as the claimed advance in imaging technology, with some interesting technical notes relevant to the field as part of that validation.

However, further, more minor, edits should be made before publication:

We thank the reviewer for his/her positive feedback on the revised version, and for the complementary minors comments that have been all addressed.

- Fig. 5:

-- The ‘holes’ that are claimed may be 18 nm fenestrations are not yet clear in the figure. These should be highlighted in the images. If they cannot be convincingly pointed out, the discussion (lines 248-250) should be modified.

These images have been made bigger for more clarity, and cavities are now pointed out by arrows.

- Confidence intervals:

-- Towards the start of the paragraph starting on line 157, a similar mean +- confidence interval for localization precision with the Gaussian beam would also be informative for comparison.

Presumably, the confidence interval would be considerably larger for the Gaussian beam than with ASTER.

-- These confidence intervals need to be explicitly defined at some point (e.g. as mean +/- s.d.)

We now added the confidence interval for the gaussian beam (l 162) and a definition of these confidence interval where they first appear.

- Ref. 48 is incomplete.

The reference has been completed

- Suppl. Note 1:

-- Include references in the text and table.

-- Explain why 16 lines is a useful number for ASTER in this comparison (e.g. the size the resulting FOV, using a Gaussian beam with sigma = x nm, and a line spacing of x sigma).

Supplementary Note 1 has been modified to include all these suggestions.

- Suppl. Note 2:

-- Define n_minline and T_min explicitly, for clarity.

-- 'scans lines' should be changed for clarity, maybe to 'line scans'.

-- Why does D need to be the longest length – why should it not be the shortest length? It would be good to explain this, or just settle for 'the length dimension D', rather than 'its longest length'.

n_minline and T_min are now described more explicitly, and "scans lines" replaced, as suggested.

Indeed, D can be chosen as the shortest length, while the scanned line will have a length similar to the longest length. This allows for fastest field generation and has been modified.

- Line 137:

-- 'spinning TIRF would prevent further quantification': This should be edited to specifically refer to the larger FOV allowed by ASTER, rather than implying more general problems with spinning TIRF.

The sentence (line 138) has been modified accordingly

- Suppl. Note 3:

-- Line 2 and 5: Should this refer to Suppl. Fig. 5, not Suppl. Fig. 3?

Yes, indeed this reference has been corrected.

-- Line 1: The mean TIRF penetration depth is 117 nm in Suppl. Fig. 5 and the main text. Should this replace the 115 nm currently given in Suppl. Note 3? If not, where does it come from?

Indeed the depth has been changed to 117 nm, which is the value of the TIRF sectioning depth we calculated, and a new figure supplementary figure 5 has been introduced, to add more details to the former Supplementary Figure 5 (now supplementary figure 6 in this new version)

-- The sentences discussing the precision limit due to pixel size should be condensed for clarity.

This has now been modified.

- Suppl. Fig. 7

-- There seems to be no panel (b). It still makes sense, and it is up to you and the editors whether that needs changing.

-- It is not currently obvious that laser speckle is specifically seen in (c), although this is implied by 'interference speckles' in main text line 134. Either the presence of laser speckle should be discussed in Suppl. Fig. 7 if the authors wish to highlight it in Suppl. Fig. 7c, or more generally, the main text could be changed to 'interference patterns.'

Mention to a) as been removed in Suppl Fig . 7

As the reviewer points out interference patterns are seen in Suppl. Fig. (c) but not laser speckle, which is generally in the order of the pixel size of 100nm and thus may not be observable. This has been corrected in the main text.

- Ref. 34 is incomplete.

Ref is now complete.

- The statement that hollowness is 'rather independent of the size of the cluster' in Methods.

-- The authors suggested two ways to edit this in their response to my comment. I recommend the option of removing that statement/information, unless it is to be given in the main text and justified in new supplementary information. Use of the image sent with the authors' response may introduce further questions if used as is.

This has been removed as suggested

- Fig. 5:

-- In the legend for (d), 'inlet' should read 'inset'.

-- The line profile in (e): apart from the shape of the line, details of that plot are poorly legible.

Figure 5 and legends have been changed to include these suggestions and panel c and e increase in size.

- Line 287:

-- 'Better resolution' should instead refer to localization precision, concerning the result for the 70 um FOV.

This has been changed accordingly (line 289)

REVIEWERS' COMMENTS

Reviewer #2 (Remarks to the Author):

There are a couple of things left to address in the revisions, but otherwise, I recommend publication:

The legend to new Suppl. Fig. 5 is incomplete. I am sure the rest of the sentence would be interesting!

The new arrows that have been included in Fig. 5c to point out smaller holes (discussed in lines 265-267) are not explained in the legend to Fig. 5.

Detailed responses to reviewers

Revised version manuscript NCOMMS-20-18746B

We would like to thank the reviewers for their thorough assessment of our revised manuscript and their useful comments. We address here all the remaining comments

REVIEWERS' COMMENTS

Reviewer #2 (Remarks to the Author):

There are a couple of things left to address in the revisions, but otherwise, I recommend publication:

The legend to new Suppl. Fig. 5 is incomplete. I am sure the rest of the sentence would be interesting!

We apologize , it apparently happened during the pdf conversion as the doc was ok.

The new arrows that have been included in Fig. 5c to point out smaller holes (discussed in lines 265-267) are not explained in the legend to Fig. 5.

We thank the reviewer for this suggestion, we have included a description within the legend, that was indeed so far only present in the text.